# PUM-Net: Plastic Unified Memory Network with Associative Interaction for Long-Context State Space Models

## Abstract

Recent Mamba-family State Space Models (SSMs) leverage linear recurrent dynamics to achieve efficiency in long-sequence modeling. However, their exponential decay kernels inevitably cause rapid forgetting, making it difficult to recall information once sequence lengths far exceed the training horizon. Inspired by advances in Transformer-based architectures such as Native Sparse Attention (NSA), which employ internal chunk memory to alleviate long-term forgetting, we extend Mamba with a chunk-wise internal long-term memory that improves the retrieval of distant context. While this enhances in-context recall, a more fundamental challenge for long-context models is the ability to access and integrate vast external world knowledge without compromising efficiency. Existing retrieval-augmented generation (RAG) approaches attempt to address this by appending retrieved documents to queries, which substantially increases training cost and fails to fully integrate internal and external memory representations. To overcome these limitations, we propose the Plastic Unified Memory Network (PUM-Net), a unified dual-memory architecture that, for the first time, enables joint pre-training over both dynamic internal memory and static, pre-encoded external knowledge. This plastic unification allows external memory to refine internal states during training, enabling bidirectional interaction without inflating sequence length, thereby supporting more effective long-context modeling and achieving substantial relative improvements across challenging benchmarks.

## 1 Introduction

The ability to process and reason over long-context information is a critical frontier for modern Large Language Models (LLMs). While Transformer-based architectures (Vaswani et al., 2017) have demonstrated remarkable capabilities, their self-attention mechanism incurs a computational and memory cost that scales quadratically with sequence length (Tay et al., 2022). This limitation presents a significant barrier to modeling extensive documents, lengthy conversations, or entire codebases. In response, a new class of architectures with near-linear complexity has emerged, most notably State Space Models (SSMs) like the Mamba architecture (Gu & Dao, 2023), which have become a promising alternative for efficient long-sequence modeling.

However, the efficiency of SSMs comes at a cost. Their core mechanism, which relies on an exponential decay kernel to compress past information into a fixed-size state, inevitably leads to rapid information forgetting and challenges in length extrapolation (Yen et al., 2024; Chen et al., 2024). This makes it difficult for the model to accurately recall specific details from distant parts of a sequence, a phenomenon often termed the "lost in the middle" problem (Liu et al., 2023a). To mitigate this, one line of research has focused on enhancing the model's internal memory. This approach is inspired by advances in efficient Transformers that employ chunking or sparse attention mechanisms to preserve long-term context, such as Longformer (Beltagy et al., 2020), BigBird (Zaheer et al., 2020), and Native Sparse Attention (NSA) (Yuan et al., 2025). Following this direction, we first extend Mamba with a chunk-wise internal memory, which substantially improves its ability to retrieve information from within the input sequence.

While strengthening internal recall is a necessary step, a more fundamental challenge is integrating vast external world knowledge without compromising efficiency. The conventional paradigm for this is Retrieval-Augmented Generation (RAG) (Lewis et al., 2020), which appends retrieved documents to the model's input. Although effective, this method is computationally expensive, as it directly inflates the sequence length (Li et al., 2025), and often results in a superficial concatenation rather than a deep fusion of knowledge (Asai et al., 2024). Pioneering work has explored pre-training models to learn to retrieve, such as REALM (Guu et al., 2020) and RETRO (Borgeaud et al., 2022), but these still rely on processing raw text during training and inference. The challenge of integrating *pre-encoded external knowledge*—such as knowledge graph embeddings or document vectors—directly into a model's core architecture during pre-training has remained largely unaddressed.

To address these limitations, we propose the Plastic Unified Memory Network (PUM-Net). We argue that a truly effective long-context model must not only maintain a robust internal memory but also seamlessly fuse it with a vast, static external knowledge base. PUM-Net is designed to achieve this through a novel joint pre-training methodology. Our contributions are threefold:

1. **For Internal Memory:** We introduce a chunk-wise long-term memory mechanism for Mamba-based SSMs. This architectural enhancement significantly mitigates the inherent forgetting problem for information within the input context and improves in-sequence recall over long distances.

2. **For External Memory:** We propose a novel pre-training methodology to deeply integrate a static, pre-encoded external knowledge base into the model's learning process. This is the first demonstration of how SSMs can be trained to fuse external knowledge without the costly concatenation of raw text.

3. **A Unified Architecture and System:** We combine these advancements in our Plastic Unified Memory Network (PUM-Net). We demonstrate that the resulting synergy from the deep integration of both internal and external memory systems leads to substantial improvements on challenging long-context benchmarks.

## 2 RELATED WORK

### 2.1 LONG-CONTEXT MODELING IN STATE SPACE MODELS

State Space Models (SSMs), adapted from control theory, have recently become competitive architectures for sequence modeling. The Structured State Space Sequence Model (S4) (Gu et al., 2022) introduced a discrete-time representation enabling parallel convolutional training and recurrent inference. However, early SSMs were largely time-invariant, limiting their ability to capture content-dependent dynamics.

Mamba (Gu & Dao, 2023) advanced this line by introducing input-dependent state transitions, allowing selective remembering and forgetting, and achieving near-Transformer performance with linear complexity. Nevertheless, their reliance on exponential decay kernels restricts retention of very long dependencies, as shown in later analyses (Chen et al., 2024; Yen et al., 2024). Our work addresses this by augmenting Mamba with explicit long-term memory to mitigate natural forgetting.

### 2.2 MEMORY MECHANISMS IN LANGUAGE MODELS

The Transformer (Vaswani et al., 2017) faces quadratic complexity, motivating efficient attention variants. Longformer (Beltagy et al., 2020) and BigBird (Zaheer et al., 2020) approximate full attention with sparse patterns. Transformer-XL (Dai et al., 2019) introduced recurrence to propagate context beyond fixed windows. More recently, Native Sparse Attention (NSA) (Yuan et al., 2025) proposed chunking and hierarchical mechanisms for structured memory. Inspired by these, our internal memory design adapts block-based memory concepts from Transformers to the recurrent dynamics of SSMs.

### 2.3 RETRIEVAL AND KNOWLEDGE INTEGRATION

Retrieval-Augmented Generation (RAG) (Lewis et al., 2020) appends retrieved passages to prompts, but this inflates sequence length (Li et al., 2025) and often yields shallow fusion of knowledge

(Asai et al., 2024). To improve integration, models like REALM (Guu et al., 2020) and RETRO (Borgeaud et al., 2022) jointly pre-trained retrievers with LMs, though still operating on raw text chunks. Other approaches (Liu et al., 2023b) explored injecting external knowledge but retained text-based overhead.

A less explored direction is leveraging pre-encoded, static knowledge (e.g., embeddings) directly during training. PUM-Net is the first to demonstrate this for SSMs, enabling deep fusion between internal dynamic states and static external memory, avoiding the inefficiencies of text-based retrieval.

## 3 THE PUM-NET ARCHITECTURE

We propose the Plastic Unified Memory Network (PUM-Net), which addresses the limitations of State Space Models (SSMs) in capturing long-range dependencies and grounding reasoning in external knowledge. PUM-Net augments a standard SSM with a dual-memory system: (i) a dynamic internal memory that encodes input sequences as chunked representations, and (ii) a static external memory of pre-encoded knowledge. A learned interaction module retrieves and fuses both memories to enrich the recurrent state. We next formalize the memory design, describe the fusion mechanism, and present the joint pre-training and inference strategy. For clarity and reproducibility, a detailed guide to the mathematical notation used throughout this section, including indices for batches, chunks, and time steps, is provided in Appendix A.

### 3.1 THE DUAL-MEMORY SYSTEM

The core of PUM-Net is its dual-memory system, which consists of a static external memory for world knowledge and a dynamic internal memory for session-specific context. Both memories adhere to a unified key-value structure. Further details on the construction of the internal and external memories, along with key hyperparameters such as the memory chunk size and the number of top-k memories selected for fusion, are provided in Appendix B.

#### 3.1.1 EXTERNAL MEMORY: STATIC KNOWLEDGE CORPUS

The external memory, $\mathcal{M}_{\text{ext}}$, is a static, pre-computed key-value store derived from a large corpus $\mathcal{D}_{\text{ext}} = \{d_1, \ldots, d_{N_{\text{ext}}}\}$.

**Key-Value Generation.** For each passage $d_i$, we generate a semantic key $\mathbf{k}_{\text{ext},i}$ and a state value $\mathbf{s}_{\text{ext},i}$. The key is produced by a frozen sentence embedding model, $\mathcal{E}_{\text{key}}$; in this paper, all-MiniLM-L6-v2[1]. . The state value is the final hidden state computed by a pre-trained Mamba-2.7B[2] model with fixed parameters $\theta_{\text{frozen}}$.

$$\mathbf{k}_{\text{ext},i} = \mathcal{E}_{\text{key}}(d_i) \in \mathbb{R}^{d_{\text{key}}}, \quad \mathbf{s}_{\text{ext},i} = f_{\theta_{\text{frozen}}}(d_i) \in \mathbb{R}^{d_{\text{state}}}. \tag{1}$$

The resulting memory is a set of tuples $\mathcal{M}_{\text{ext}} = \{(\mathbf{k}_{\text{ext},i}, \mathbf{s}_{\text{ext},i})\}_{i=1}^{N_{\text{ext}}}$.

**Indexing.** The keys $\{\mathbf{k}_{\text{ext},i}\}$ are organized via an Approximate Nearest Neighbor (ANN) index $\mathcal{I}_{\text{ext}}$. We use an inverted file index (IVF) with $K$ centroids $\{\mathbf{c}_k\}_{k=1}^{K}$. Given a query $\mathbf{q} \in \mathbb{R}^{d_{\text{key}}}$, retrieval is restricted to a candidate set $\mathcal{K}_{\text{cand}}(\mathbf{q})$ comprising keys from nearby clusters. Top-$N$ retrieval is then:

$$\text{ANN-Search}(\mathbf{q}, N) = \underset{\mathbf{k}_{\text{ext},i} \in \mathcal{K}_{\text{cand}}(\mathbf{q})}{\arg\min}^{N} \|\mathbf{q} - \mathbf{k}_{\text{ext},i}\|_2. \tag{2}$$

#### 3.1.2 INTERNAL MEMORY: DYNAMIC CONTEXTUAL REPRESENTATION

The internal memory, $\mathcal{M}_{\text{int}}$, is constructed on-the-fly for each input sequence $\mathbf{U}$. Unlike the static external memory, its keys are not indexed into a persistent structure to avoid prohibitive computational overhead during training.

---

[1]https://huggingface.co/sentence-transformers/all-MiniLM-L6-v2

[2]https://huggingface.co/fla-hub/mamba-2.7B-100B

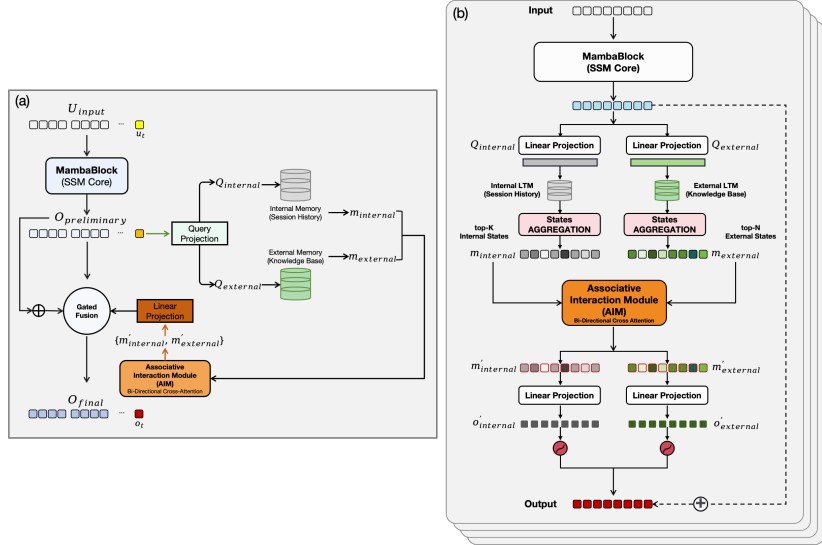

Figure 1: Illustration of the proposed PUM-Net. (a) The overall workflow. (b) The block design.

**Chunking and Key-Value Generation.** The input $\mathbf{U}$ is partitioned into $M$ chunks of length $L_c$, $\{\mathbf{u}_j\}_{j=1}^M$. For each chunk, a key-value pair is generated. The key $\mathbf{k}_{\text{int},j} \in \mathbb{R}^{d_{\text{key}}}$ is computed using the shared encoder $\mathcal{E}_{\text{key}}$. The state value $\mathbf{s}_{\text{int},j} \in \mathbb{R}^{d_{\text{state}}}$ is the final hidden state computed by the learnable PUM-Net backbone $f_\theta$:

$$\mathbf{s}_{\text{int},j} = f_\theta(\mathbf{u}_j). \tag{3}$$

**Parallel Computation and Caching.** To compute all state values efficiently for a training batch of $B$ sequences, the $B \times M$ chunks are processed in a single forward pass. Their token embeddings form a tensor $\mathbf{E}_{\text{batch}} \in \mathbb{R}^{(B \times M) \times L_c \times d_{\text{model}}}$. The backbone $f_\theta$ operates on this tensor to produce the final states $\mathbf{X}_{\text{batch}} \in \mathbb{R}^{(B \times M) \times d_{\text{state}}}$, from which each individual state $\mathbf{s}_{\text{int},j}^{(b)}$ is extracted. For each sequence in the batch, the collection of key-value pairs is held in a temporary cache for the duration of the forward pass:

$$\mathcal{M}_{\text{int}}^{(b)} = \{(\mathbf{k}_{\text{int},j}^{(b)}, \mathbf{s}_{\text{int},j}^{(b)})\}_{j=1}^M. \tag{4}$$

Retrieval from this small, session-specific cache is performed via efficient brute-force similarity search, as detailed in the following section.

Fig. 1 illustrates the proposed PUM-Net, where (a) shows the overall workflow and (b) details the block design.

## 3.2 Training Paradigm: Staged Parallel Computation

PUM-Net training is complicated by recurrent dependencies, where computations at step $t$ rely on retrievals conditioned on the same state. To address this and enable efficient accelerator utilization, we introduce **Staged Parallel Computation**, which decomposes the forward pass into two parallelizable stages, preserving end-to-end differentiability and gradient flow.

### 3.2.1 Stage 1: Preliminary State Scan

First, a preliminary parallel scan is performed over the input embeddings $\mathbf{E} \in \mathbb{R}^{B \times L \times d_{\text{model}}}$ to generate context-aware representations. The backbone SSM, parameterized by $\theta$, computes:

$$\mathbf{O}_{\text{prelim}} = f_\theta(\mathbf{E}) \in \mathbb{R}^{B \times L \times d_{\text{model}}}. \tag{5}$$

Each vector $\mathbf{o}_{\text{prelim},t}^{(b)}$ in this tensor serves as a contextualized basis for memory querying.

### 3.2.2 STAGE 2: PARALLEL RETRIEVAL AND ASSOCIATIVE INTERACTION

**a) Parallel Query Generation.** The preliminary states are projected into query vectors for both memory systems via learnable affine transformations:

$$\mathbf{Q}_{\text{int}} = \mathbf{O}_{\text{prelim}}\mathbf{W}_{q,\text{int}} \in \mathbb{R}^{B \times L \times d_{\text{key}}}, \quad \mathbf{Q}_{\text{ext}} = \mathbf{O}_{\text{prelim}}\mathbf{W}_{q,\text{ext}} \in \mathbb{R}^{B \times L \times d_{\text{key}}}. \quad (6)$$

**b) Batched Dual-Memory Retrieval.** At each time step $t$, we retrieve the top-$k$ most relevant states from both memories in parallel. For the static external memory, we query the pre-built ANN index $\mathcal{I}_{\text{ext}}$ to efficiently retrieve the top-$k$ candidates for each query $\mathbf{q}_{\text{ext},t}^{(b)}$:

$$\mathcal{S}_{\text{ext},t}^{(b)} = \{\mathbf{s}_{\text{ext},i}\}_{i \in \text{ANN-Search}(\mathbf{q}_{\text{ext},t}^{(b)},k)}. \quad (7)$$

For the dynamic internal memory, an exhaustive search is performed via a single, massively parallel computation. The dot-product similarities between all query vectors and all internal memory keys in the batch are computed at once using efficient matrix multiplication. From the resulting similarity scores, we retrieve the top-$k$ states for each time step:

$$\mathcal{S}_{\text{int},t}^{(b)} = \{\mathbf{s}_{\text{int},j}^{(b)}\}_{j \in \text{Top-k-Indices}(\mathbf{q}_{\text{int},t}^{(b)}, \{\mathbf{k}_{\text{int},j'}^{(b)}\}_{j'=1}^{M})}. \quad (8)$$

**c) Bi-Directional Cross-Memory Interaction** The Associative Interaction Module (AIM) processes the retrieved states through a sequence of similarity-weighted aggregation, bi-directional cross-attention refinement.

SIMILARITY-WEIGHTED AGGREGATION. The retrieved states from each memory, $\mathcal{S}_{\text{int},t}^{(b)}$ and $\mathcal{S}_{\text{ext},t}^{(b)}$, are aggregated into single context vectors using a standard attention mechanism. Attention weights ($\alpha$) are computed via scaled dot-product similarity between the query and the retrieved keys. These weights are then used to produce a weighted sum of the retrieved states, yielding the final context vectors $\mathbf{m}_{\text{int},t}^{(b)}$ and (analogously) $\mathbf{m}_{\text{ext},t}^{(b)}$:

$$\alpha_{t,j}^{(b)} = \text{softmax}_j\left(\frac{(\mathbf{q}_{\text{int},t}^{(b)})^{\top}\mathbf{k}_{\text{int},j}^{(b)}}{\sqrt{d_{\text{key}}}}\right), \quad \mathbf{m}_{\text{int},t}^{(b)} = \sum_j \alpha_{t,j}^{(b)}\mathbf{s}_{\text{int},j}^{(b)}. \quad (9)$$

BI-DIRECTIONAL CROSS-ATTENTION REFINEMENT. To foster deep integration and mutual refinement, the aggregated context vectors interact through a bi-directional cross-attention mechanism. First, external knowledge enriches the internal context representation:

$$\mathbf{Q}_1 = \mathbf{m}_{\text{int},t}^{(b)}\mathbf{W}_{Q_1}, \quad \mathbf{K}_1, \mathbf{V}_1 = \mathbf{m}_{\text{ext},t}^{(b)}\mathbf{W}_{K_1}, \mathbf{m}_{\text{ext},t}^{(b)}\mathbf{W}_{V_1}, \quad (10)$$

$$\mathbf{m}_{\text{int},t}'^{(b)} = \text{LayerNorm}(\mathbf{m}_{\text{int},t}^{(b)} + \text{Attention}(\mathbf{Q}_1, \mathbf{K}_1, \mathbf{V}_1)). \quad (11)$$

Concurrently, the internal context grounds the external knowledge, producing a session-specific external representation:

$$\mathbf{Q}_2 = \mathbf{m}_{\text{ext},t}^{(b)}\mathbf{W}_{Q_2}, \quad \mathbf{K}_2, \mathbf{V}_2 = \mathbf{m}_{\text{int},t}^{(b)}\mathbf{W}_{K_2}, \mathbf{m}_{\text{int},t}^{(b)}\mathbf{W}_{V_2}, \quad (12)$$

$$\mathbf{m}_{\text{ext},t}'^{(b)} = \text{LayerNorm}(\mathbf{m}_{\text{ext},t}^{(b)} + \text{Attention}(\mathbf{Q}_2, \mathbf{K}_2, \mathbf{V}_2)). \quad (13)$$

This reciprocal process produces two mutually-informed representations, $\mathbf{m}_{\text{int},t}'^{(b)}$ and $\mathbf{m}_{\text{ext},t}'^{(b)}$, which are then passed to the fusion stage.

**d) Gated Fusion for Final Output Computation.** The final model output is computed in a single efficient step. The refined memory representations are first projected from the state space $\mathbb{R}^{d_{\text{state}}}$ to the model's output space $\mathbb{R}^{d_{\text{model}}}$. A sophisticated gating mechanism then adaptively integrates these memory contributions into the preliminary outputs from Stage 1. For each timestep $t$ and batch element $b$:

$$\mathbf{o}_{\text{int},t}^{(b)} = \mathbf{m}_{\text{int},t}'^{(b)}\mathbf{W}_{m,\text{int}}, \quad \mathbf{o}_{\text{ext},t}^{(b)} = \mathbf{m}_{\text{ext},t}'^{(b)}\mathbf{W}_{m,\text{ext}}, \quad (14)$$

$$\mathbf{g}_{\text{int},t}^{(b)} = \sigma([\mathbf{o}_{\text{prelim},t}^{(b)}; \mathbf{o}_{\text{int},t}^{(b)}]\mathbf{W}_{g,\text{int}}), \quad \mathbf{g}_{\text{ext},t}^{(b)} = \sigma([\mathbf{o}_{\text{prelim},t}^{(b)}; \mathbf{o}_{\text{ext},t}^{(b)}]\mathbf{W}_{g,\text{ext}}), \quad (15)$$

$$\mathbf{o}_{\text{final},t}^{(b)} = \mathbf{o}_{\text{prelim},t}^{(b)} + \mathbf{g}_{\text{int},t}^{(b)} \odot \mathbf{o}_{\text{int},t}^{(b)} + \mathbf{g}_{\text{ext},t}^{(b)} \odot \mathbf{o}_{\text{ext},t}^{(b)}. \quad (16)$$

Here, $[\cdot;\cdot]$ denotes concatenation, and all $\mathbf{W}$ matrices are learnable parameters. This formulation bypasses a second SSM scan, directly yielding the final output tensor $\mathbf{O}_{\text{final}} \in \mathbb{R}^{B \times L \times d_{\text{model}}}$ for prediction.

### 3.2.3 TRAINING OBJECTIVE

The model is trained end-to-end using a standard autoregressive language modeling objective. The final outputs $\mathbf{O}_{\text{final}}$ are projected to the vocabulary space via a linear layer, $\mathbf{W}_{\text{vocab}}$, to produce logits $\mathbf{L} \in \mathbb{R}^{B \times L \times |\mathcal{V}|}$. The training loss is the cross-entropy between the predicted logits and the ground-truth target tokens:

$$\mathcal{L}(\theta) = \sum_{t=1}^{L} \text{CrossEntropy}(\text{softmax}(\mathbf{L}_t), \text{target}_t). \tag{17}$$

The gradient is backpropagated through the entire staged computation graph to update all learnable parameters $\theta$.

### 3.2.4 INFERENCE PROCEDURE

Inference mirrors the training pipeline with two differences: (i) the static external memory $\mathcal{M}_{\text{ext}}$ and its ANN index $\mathcal{I}_{\text{ext}}$ are already built offline and never reconstructed at test time; and (ii) after a one-time forward scan over the input *query* sequence $U_q$ to obtain $O_{\text{prelim},1:|U_q|}$ and the final recurrent state $x_{|U_q|}$, answer tokens are generated autoregressively. At each decoding step, we reuse the *trained* query generators ($W_{q,\text{int}}$, $W_{q,\text{ext}}$), dual-memory retrieval (internal via in-session brute-force over $\mathcal{M}_{\text{int}}(U_q)$; external via $\mathcal{I}_{\text{ext}}$), similarity-weighted aggregation, bi-directional cross-attention refinement (AIM), linear projections ($W_{m,*}$), and gating ($W_{g,*}$) to fuse memory signals into $o_{\text{final},t}$ for prediction, thereby incurring only a small, fixed per-token overhead without any additional SSM scan. Due to space constraints, detailed pseudocode for our staged parallel training and autoregressive inference procedures is provided in Appendix C.

## 4 EXPERIMENTS

We conduct a series of experiments to validate the effectiveness of the PUM-Net architecture. Our evaluation is designed to answer two primary questions: (1) Does PUM-Net outperform standard State Space Model (SSM) baselines in long-range modeling tasks? (2) What are the individual contributions of its internal and external memory components? We address these questions through comprehensive benchmarks evaluating perplexity, in-context recall, question answering, and computational efficiency. A discussion on why our experiments do not include a direct comparison with traditional Retrieval-Augmented Generation (RAG) methods is provided in Appendix H.

### 4.1 LONG-RANGE LANGUAGE MODELING

**Setup.** We first evaluate PUM-Net on the task of long-range language modeling, using three widely recognized datasets: **PG-19** (Rae et al., 2019), **ProofPile** (Azerbayev et al., 2023), and **CodeParrot** (Thomas Wolf & Zebaze, 2023). All models are trained with a 4k token context and evaluated for perplexity (PPL) on sequences up to 64k tokens to test extrapolation capabilities. Further details on the data set split, each task's external memory source, and training protocol are provided in the Appendix D.

**Results.** Figure 2 shows that the standard Mamba-130M baseline's performance degrades sharply beyond its training length. In contrast, **PUM-Net (w/o ex)**, which isolates our internal memory mechanism, maintains significantly lower perplexity, demonstrating its effectiveness at mitigating information decay. The full **PUM-Net** model achieves the best results across all contexts. The benefit of its external memory is especially pronounced on CodeParrot, underscoring its value for knowledge-intensive domains. As shown in the training loss curves (Figure 3), PUM-Net also exhibits a more favorable optimization landscape, suggesting convergence to a better local minimum.

Table 1 benchmarks PUM-Net against vanilla Mamba and several Mamba variants augmented with attention mechanisms like Sliding Window Attention (SWA) and Native Sparse Attention (NSA).

Table 1: Perplexity at evaluation lengths up to 64k after fine-tuning on 4k. Baselines include Transformer and Mamba variants with SWA and NSA. Our method, PUM-Net, is shown in two forms: (w/o ex) without external memory and (full) with external memory.

| Dataset | Length | Transformer$_{\text{full\_attn}}$ | Mamba-130M | Mamba w/ SWA$_{\text{rope}}$ | Mamba w/ NSA | PUM-Net (w/o ex) | PUM-Net (full) |
|---|---|---|---|---|---|---|---|
| **PG-19** | 4k | 16.40 | 16.69 | 16.45 | 16.38 | 16.12 | **15.96** |
| | 8k | 17.10 | 17.35 | 17.01 | 16.92 | 16.77 | **16.63** |
| | 16k | 520.0 | 17.79 | 17.55 | 17.43 | 16.98 | **16.86** |
| | 32k | 1800.0 | 26.54 | 25.82 | 25.60 | 22.32 | **22.23** |
| | 64k | $1.0 \times 10^5$ | 4352.97 | 3890.24 | 3725.50 | 619.33 | **411.29** |
| **ProofPile** | 4k | 4.40 | 4.52 | 4.38 | 4.31 | 4.01 | **3.53** |
| | 8k | 4.55 | 4.60 | 4.53 | 4.49 | 4.17 | **3.61** |
| | 16k | 150.0 | 5.37 | 5.21 | 5.14 | 4.84 | **4.40** |
| | 32k | 1200.0 | 28.02 | 27.10 | 26.83 | 27.54 | **26.50** |
| | 64k | $5.0 \times 10^4$ | 76645.95 | 69220.00 | 67310.50 | 2425.50 | **1265.57** |
| **CodeParrot** | 4k | 4.55 | 4.61 | 4.50 | 4.48 | 4.47 | **1.11** |
| | 8k | 5.20 | 5.34 | 5.22 | 5.19 | 5.17 | **3.18** |
| | 16k | 120.0 | 7.31 | 7.20 | 7.15 | 7.18 | **4.25** |
| | 32k | 1500.0 | 26.01 | 25.62 | 25.50 | 25.14 | **22.50** |
| | 64k | $1.0 \times 10^9$ | $1.53 \times 10^{10}$ | $1.28 \times 10^{10}$ | $1.19 \times 10^{10}$ | $2.14 \times 10^9$ | **6288** |

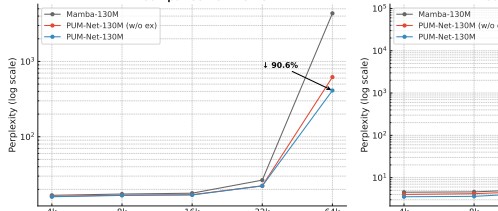
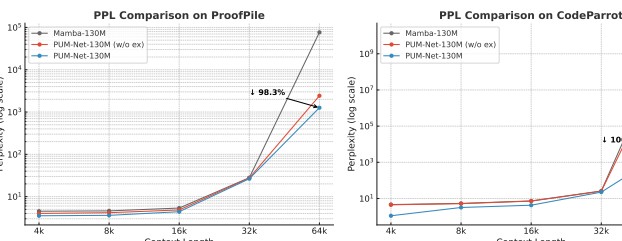

Figure 2: Perplexity (PPL) comparison on the PG-19, ProofPile, and CodeParrot test sets. All models are trained on a 4k context length and evaluated on lengths up to 64k. Lower PPL indicates better performance. PUM-Net consistently outperforms the Mamba baseline, with the performance gap widening dramatically at longer contexts. The addition of external memory provides a significant further boost, especially on the knowledge-intensive CodeParrot dataset.

For fairness, all models are implemented based on a 130M parameter budget (see Appendix D.1 for details). The results show that PUM-Net consistently outperforms all baselines. This strong performance against the NSA-augmented model is particularly significant, as our architecture was inspired by similar principles of efficient information routing. The full **PUM-Net** model achieves state-of-the-art results, with its external memory showing a pronounced advantage on the knowledge-intensive CodeParrot dataset.

### 4.2 PASSKEY RETRIEVAL: EVALUATING INTERNAL MEMORY

**Setup.** To specifically isolate the internal memory component, we use the passkey retrieval task, a synthetic "needle-in-a-haystack" benchmark that requires verbatim recall of information from a long sequence. Since this task relies purely on in-context information, we evaluate **PUM-Net (w/o ex)** against several Mamba-family baselines. Setup specifics are available in Appendix E.

**Results.** Figure 4 shows that while baseline models fail as context extends beyond their training length, **PUM-Net (w/o ex)** maintains near-perfect retrieval accuracy up to 64k tokens. This result provides strong evidence that our dynamic internal memory acts as a reliable long-term buffer, overcoming the typical forgetting problem of SSMs.

### 4.3 PERFORMANCE ON LONG-CONTEXT QUESTION ANSWERING

**Setup.** We evaluate PUM-Net on a suite of question-answering tasks from the **LongBench** benchmark (Bai et al., 2023). These tasks require models to find and reason over information in long documents, testing the synergy between internal and external memory. We compare the full **PUM-**

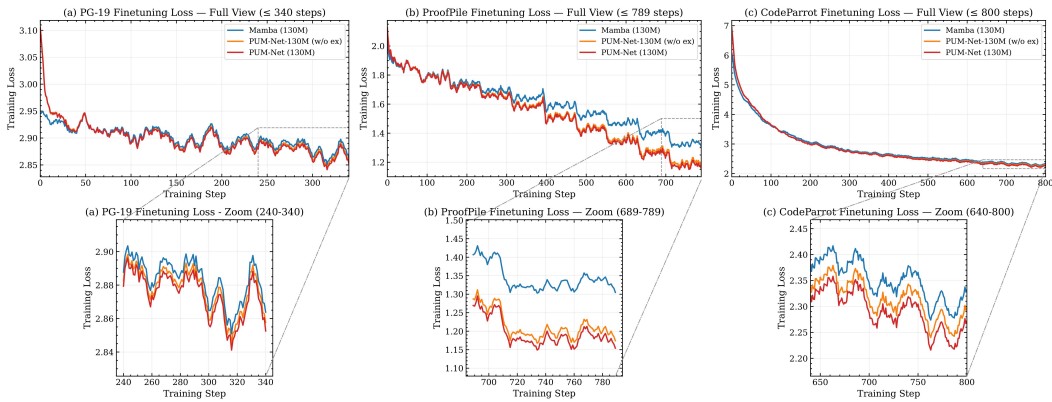

Figure 3: Training loss curves for PG-19, ProofPile, and CodeParrot finetuning. Each column shows the full training run and a zoomed-in view of the final training steps. PUM-Net exhibits lower loss throughout training and demonstrates stronger late-stage advantages compared to both vanilla Mamba and the variant equipped only with internal memory, indicating a superior convergence point.

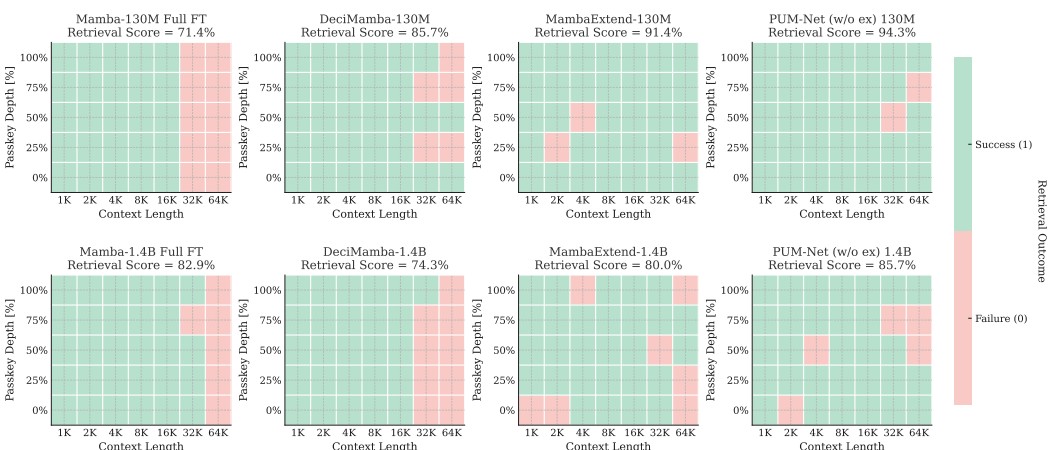

Figure 4: Passkey retrieval performance comparison. Baseline models, fine-tuned on 4k context length samples, show significant degradation on longer contexts. In contrast, our PUM-Net (w/o ex) model demonstrates robust performance, maintaining high retrieval accuracy up to 64k tokens.

**Net-2.8B** against a fine-tuned **Mamba-2.8B** and our **PUM-Net (w/o ex)** ablation, measuring performance with F1 score. Experiment details and external memory construction for this task are available in the appendix F.

**Results.** The results in Table 2 demonstrate PUM-Net's superior reasoning capabilities. On all tasks, **PUM-Net (w/o ex)** outperforms the Mamba baseline, confirming the benefit of the improved internal memory for downstream tasks. The full **PUM-Net** model consistently yields the best scores, showing that the external memory provides a crucial advantage for synthesizing information to answer complex questions.

## 4.4 TRAINING AND INFERENCE EFFICIENCY

**Setup.** Finally, we evaluate the computational efficiency of PUM-Net. We measure throughput (forward/backward pass time) and peak memory usage against highly optimized Transformer base-

Table 2: Results on LongBench QA tasks after finetuning. We compare Mamba-2.8b baseline, PUM-Net without external memory, and PUM-Net with external memory. Adding external memory provides consistent further improvements, highlighting its benefits for QA tasks. LB = LongBench score, and 'N/A' means that the task does not exist in LongBench-E.

| Type (Metric) | Benchmark | Avg Len | Mamba-2.8b (finetuned) | | | | PUM-Net (w/o ex) | | | | PUM-Net (full) | | | |
|---|---|---|---|---|---|---|---|---|---|---|---|---|---|---|
| | | | 0-4k | 4-8k | 8k+ | LB | 0-4k | 4-8k | 8k+ | LB | 0-4k | 4-8k | 8k+ | LB |
| MultiDoc-QA (F1) | 2wikimqa | 4887 | 8.47 | 2.34 | 1.18 | 4.53 | 10.72 | 5.48 | 2.93 | 9.58 | **12.42** | **6.84** | **3.63** | **11.46** |
| MultiDoc-QA (F1) | Hotpotqa | 9151 | 5.77 | 2.02 | 0.53 | 2.28 | 7.08 | 3.36 | 2.03 | 4.24 | **8.88** | **4.53** | **2.71** | **5.69** |
| MultiDoc-QA (F1) | Musique | 11214 | N/A | N/A | N/A | 1.23 | N/A | N/A | N/A | 2.38 | N/A | N/A | N/A | **3.47** |
| SingleDoc-QA (F1) | NarrativeQA | 18409 | N/A | N/A | N/A | 1.27 | N/A | N/A | N/A | 2.13 | N/A | N/A | N/A | **3.08** |
| SingleDoc-QA (F1) | Qasper | 3619 | 7.52 | 5.17 | 2.14 | 6.09 | 9.14 | 8.29 | 2.73 | 9.28 | **11.18** | **10.07** | **3.63** | **11.84** |
| SingleDoc-QA (F1) | MultifieldQA | 4559 | 19.28 | 6.73 | 2.93 | 12.46 | 24.12 | 12.28 | 4.64 | 18.87 | **27.76** | **14.47** | **6.24** | **21.73** |
| Few-Shot (F1) | TriviaQA | 8209 | 11.46 | 6.38 | 4.52 | 5.83 | 14.23 | 10.07 | 7.17 | 10.36 | **16.87** | **12.82** | **9.58** | **13.69** |

lines: **Native Sparse Attention (NSA)** (Yuan et al., 2025) and **Flash-Attention** (Dao et al., 2022). The benchmark setup is detailed in Appendix G.

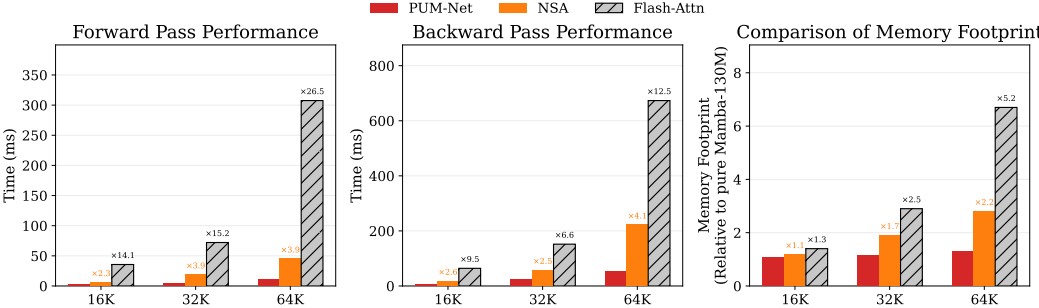

Figure 5: Efficiency comparison of PUM-Net against Native Sparse Attention (NSA) and Flash-Attention. Subplots show (from left to right): forward pass time, backward pass time, and peak memory footprint relative to a pure Mamba model. PUM-Net demonstrates superior efficiency across all metrics, with its advantages becoming more pronounced as sequence length increases.

**Results.** Figure 5 shows that PUM-Net is substantially more efficient than attention-based baselines. Its throughput is significantly higher, achieving a **3.9x speedup** over NSA and a **26.5x speedup** over Flash-Attention in the forward pass at 64k context length. Moreover, its memory footprint is markedly lower and scales more favorably. This efficiency is due to our architecture's avoidance of quadratic-cost operations, confirming that PUM-Net achieves state-of-the-art performance without sacrificing computational feasibility.

## 5 CONCLUSION

We presented the Plastic Unified Memory Network (PUM-Net), a new architecture that augments State Space Models with a unified dual-memory system, combining a dynamic chunk-wise internal memory for long-range sequence retention with a static external memory for world knowledge, trained efficiently via a staged parallel computation scheme. Empirical results on long-context language modeling benchmarks demonstrate that PUM-Net substantially improves extrapolation, alleviates in-sequence forgetting, and delivers significant gains on knowledge-intensive tasks while maintaining superior computational efficiency compared to attention-based alternatives. Despite these advances, limitations remain, including the reliance on a fixed external memory, performance degradation at extreme sequence lengths, and an evaluation primarily on text. Future work will therefore focus on developing adaptive external memory, scaling PUM-Net to even longer contexts, and extending the architecture to multi-modal reasoning. We believe this unified memory perspective provides a promising foundation for building the next generation of efficient and knowledge-grounded long-context models.

## 6 ETHICS STATEMENT

The research presented in this paper focuses on a frontier LLM model architecture, PUM-Net, and primarily utilizes established, publicly available datasets for training and evaluation. We acknowledge that, like all large language models, architectures like ours could potentially be used to generate harmful, biased, or factually incorrect content, as their behavior is a reflection of the data they are trained on. A specific consideration for our dual-memory approach is the content of the external knowledge base; the quality and neutrality of this external memory can directly influence the model's outputs. Our work is intended for research purposes to advance the understanding of efficient long-context models, and we encourage the community to pursue responsible development and deployment of such technologies.

## 7 REPRODUCIBILITY STATEMENT

We are committed to ensuring the reproducibility of our research. The source code for the PUM-Net architecture, along with the configurations for all experiments, will be made publicly available on GitHub upon publication. We plan to release our finetuned model checkpoints on the Hugging Face Hub. All experiments were conducted on publicly available benchmarks (PG-19, ProofPile, CodeParrot, LongBench). The specific data processing, sampling strategies, model implementation details, and key hyperparameters for all experiments are described in detail in the appendix of this paper. Our experiments were conducted using NVIDIA H100 GPUs.

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

# A NOTATIONAL GUIDE

We summarize the key mathematical notation used in Section 3 for clarity and ease of reference.

$L, \mathbf{U}$  Denotes the total sequence length ($L$) and a long input sequence $\mathbf{U} = (u_1, \ldots, u_L)$.

**Superscript** $(b)$  Refers to the $b$-th sequence within a batch (e.g., $\mathbf{U}^{(b)}$).

$L_c, j$  Denotes the fixed length of a chunk ($L_c$) and the index $j$ of a chunk within the sequence.

**Subscript** $t$  Refers to a specific time step or token index, where $1 \le t \le L$.

$\mathbf{q}, \mathbf{k}, \mathbf{s}, \ldots$  Bold lowercase letters represent vectors (e.g., a query vector).

$\mathbf{W}, \mathbf{O}, \mathbf{Q}, \ldots$  Bold uppercase letters represent matrices or tensors (e.g., a weight matrix).

# B MEMORY CONSTRUCTION DIAGRAM

The construction process for the static external memory and the dynamic internal memory, as detailed in the main text, is visually illustrated in Figure 6. For all experiments involving PUM-Net presented in this paper, we use a consistent set of hyperparameters for the dual-memory system. Specifically, for both the internal and external memories, the input is processed using a chunk size of 64 tokens, and the retrieval mechanism selects the top-$k$ most relevant chunks, where $k = 8$.

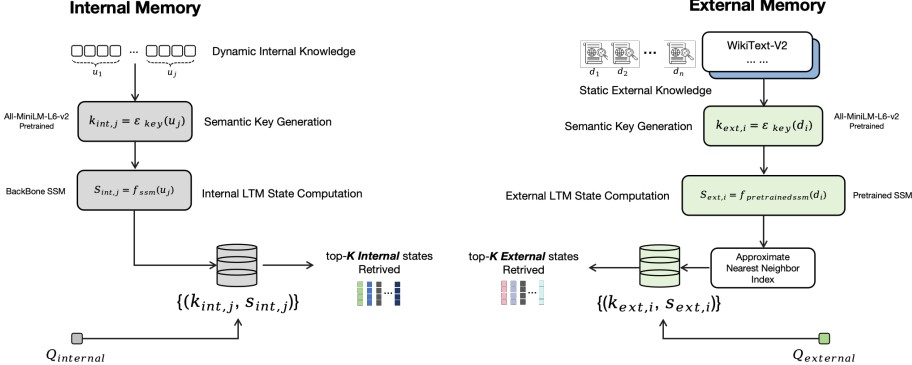

Figure 6: Overview of the construction process for the static external memory (right) and the dynamic internal memory (left). Both processes generate unified key-value pairs but utilize different data sources (an external corpus vs. the input sequence) and are performed at different stages (offline vs. dynamically).

---

**Algorithm 1** Staged Parallel Training for PUM-Net

---

**Require:** Batch embeddings $\mathbf{E} \in \mathbb{R}^{B \times L \times d_{\text{model}}}$; internal chunk keys/values $\{\mathcal{M}_{\text{int}}^{(b)}\}_{b=1}^{B}$ with $\mathcal{M}_{\text{int}}^{(b)} = \{(\mathbf{k}_{\text{int},j}^{(b)}, \mathbf{s}_{\text{int},j}^{(b)})\}_{j=1}^{M}$; external memory $\mathcal{M}_{\text{ext}} = \{(\mathbf{k}_{\text{ext},i}, \mathbf{s}_{\text{ext},i})\}_{i=1}^{N_{\text{ext}}}$ and ANN index $\mathcal{I}_{\text{ext}}$; $K, N$; parameters $\theta, W_{q,*}, W_{m,*}, W_{g,*}, W_{\text{vocab}}$.

**Ensure:** Loss $\mathcal{L}(\theta)$.

1: **Stage 1: Preliminary scan** $\quad \mathbf{O}_{\text{prelim}} \leftarrow f_\theta(\mathbf{E}) \in \mathbb{R}^{B \times L \times d_{\text{model}}}$.

2: **Stage 2a: Parallel query generation** $\quad \mathbf{Q}_{\text{int}} \leftarrow \mathbf{O}_{\text{prelim}} W_{q,\text{int}}, \ \mathbf{Q}_{\text{ext}} \leftarrow \mathbf{O}_{\text{prelim}} W_{q,\text{ext}}$.

3: **for all** $b \in \{1, \dots, B\}$ **and** $t \in \{1, \dots, L\}$ **in parallel do**

4: $\quad$ **Stage 2b: Retrieval**

5: $\quad\quad$ *Internal:* $\mathcal{S}_{\text{int},t}^{(b)} \leftarrow \{\mathbf{s}_{\text{int},j}^{(b)}\}_{j \in \text{Top-K-Indices}(\mathbf{Q}_{\text{int},t}^{(b)}, \{\mathbf{k}_{\text{int},j'}^{(b)}\}_{j'=1}^{M})}$.

6: $\quad\quad$ *External:* $\mathcal{S}_{\text{ext},t}^{(b)} \leftarrow \{\mathbf{s}_{\text{ext},i}\}_{i \in \text{ANN-Search}(\mathbf{Q}_{\text{ext},t}^{(b)}, N; \mathcal{I}_{\text{ext}})}$.

7: $\quad$ **Stage 2c: Similarity-weighted aggregation**

8: $\quad\quad$ $\alpha_{t,j}^{(b)} \leftarrow \text{softmax}_j\big((\mathbf{q}_{\text{int},t}^{(b)})^\top \mathbf{k}_{\text{int},j}^{(b)} / \sqrt{d_{\text{key}}}\big), \ \mathbf{m}_{\text{int},t}^{(b)} \leftarrow \sum_j \alpha_{t,j}^{(b)} \mathbf{s}_{\text{int},j}^{(b)}$.

9: $\quad\quad$ (Analogously obtain $\mathbf{m}_{\text{ext},t}^{(b)}$.)

10: $\quad$ **Stage 2d: Bi-directional cross-attention refinement (AIM)**

11: $\quad\quad$ $\mathbf{Q}_1 \leftarrow \mathbf{m}_{\text{int},t}^{(b)} \mathbf{W}_{Q_1}, \ \mathbf{K}_1, \mathbf{V}_1 \leftarrow \mathbf{m}_{\text{ext},t}^{(b)} \mathbf{W}_{K_1}, \mathbf{m}_{\text{ext},t}^{(b)} \mathbf{W}_{V_1}$.

12: $\quad\quad$ $\mathbf{m}_{\text{int},t}'^{(b)} \leftarrow \text{LayerNorm}\big(\mathbf{m}_{\text{int},t}^{(b)} + \text{Attention}(\mathbf{Q}_1, \mathbf{K}_1, \mathbf{V}_1)\big)$.

13: $\quad\quad$ $\mathbf{Q}_2 \leftarrow \mathbf{m}_{\text{ext},t}^{(b)} \mathbf{W}_{Q_2}, \ \mathbf{K}_2, \mathbf{V}_2 \leftarrow \mathbf{m}_{\text{int},t}^{(b)} \mathbf{W}_{K_2}, \mathbf{m}_{\text{int},t}^{(b)} \mathbf{W}_{V_2}$.

14: $\quad\quad$ $\mathbf{m}_{\text{ext},t}'^{(b)} \leftarrow \text{LayerNorm}\big(\mathbf{m}_{\text{ext},t}^{(b)} + \text{Attention}(\mathbf{Q}_2, \mathbf{K}_2, \mathbf{V}_2)\big)$.

15: $\quad$ **Stage 3: Projection and gated fusion**

16: $\quad\quad$ $\mathbf{o}_{\text{int},t}^{(b)} \leftarrow \mathbf{m}_{\text{int},t}'^{(b)} \mathbf{W}_{m,\text{int}}, \ \mathbf{o}_{\text{ext},t}^{(b)} \leftarrow \mathbf{m}_{\text{ext},t}'^{(b)} \mathbf{W}_{m,\text{ext}}$.

17: $\quad\quad$ $\mathbf{g}_{\text{int},t}^{(b)} \leftarrow \sigma([\mathbf{o}_{\text{prelim},t}^{(b)}; \mathbf{o}_{\text{int},t}^{(b)}] \mathbf{W}_{g,\text{int}}), \ \mathbf{g}_{\text{ext},t}^{(b)} \leftarrow \sigma([\mathbf{o}_{\text{prelim},t}^{(b)}; \mathbf{o}_{\text{ext},t}^{(b)}] \mathbf{W}_{g,\text{ext}})$.

18: $\quad\quad$ $\mathbf{o}_{\text{final},t}^{(b)} \leftarrow \mathbf{o}_{\text{prelim},t}^{(b)} + \mathbf{g}_{\text{int},t}^{(b)} \odot \mathbf{o}_{\text{int},t}^{(b)} + \mathbf{g}_{\text{ext},t}^{(b)} \odot \mathbf{o}_{\text{ext},t}^{(b)}$.

19: **end for**

20: **Prediction & loss** $\quad \mathbf{L}_t \leftarrow \mathbf{o}_{\text{final},t} W_{\text{vocab}}, \ \mathcal{L}(\theta) \leftarrow \sum_{t=1}^{L} \text{CrossEntropy}(\text{softmax}(\mathbf{L}_t), \text{target}_t)$.

---

## C   Pseudocode for PUM-Net Training and Inference

---

**Algorithm 2** Inference for PUM-Net

---

**Require:** Query sequence $U_q$; pre-built $\mathcal{M}_{\text{ext}}$, $\mathcal{I}_{\text{ext}}$; top-$K$, top-$N$; trained $\theta$, $W_{q,*}$, $W_{m,*}$, $W_{g,*}$, $W_{\text{vocab}}$.

**Ensure:** Generated answer tokens $y_{1:T}$.
 1: **Phase A: One-time context preparation (no generation)**
 2: Build $\mathcal{M}_{\text{int}}(U_q) = \{(\mathbf{k}_{\text{int},j}, \mathbf{s}_{\text{int},j})\}_{j=1}^{M}$ as in training (chunking $U_q$, shared $\mathcal{E}_{\text{key}}$, backbone $f_\theta$).
 3: Run a single forward scan over $U_q$ to obtain $O_{\text{prelim},1:|U_q|}$ and final state $x_{|U_q|}$.
 4: Initialize $x_0 \leftarrow x_{|U_q|}$, $y_0 = \texttt{<ANS-START>}$, $u_1 \leftarrow \text{Embed}(y_0)$.
 5: **Phase B: Autoregressive answer generation**
 6: **for** $t = 1$ to $T$ **do**
 7:     **Preliminary output:** $o_{\text{prelim},t} \leftarrow f_\theta(x_{t-1}, u_t)$.
 8:     **Dual-query generation (frozen):** $q_{\text{int},t} \leftarrow o_{\text{prelim},t} W_{q,\text{int}}$, $q_{\text{ext},t} \leftarrow o_{\text{prelim},t} W_{q,\text{ext}}$.
 9:     **Memory retrieval:** $\mathcal{S}_{\text{int},t} \leftarrow \{\mathbf{s}_{\text{int},j}\}_{j \in \text{Top-K-Indices}(q_{\text{int},t}, \{\mathbf{k}_{\text{int},j'}\}_{j'=1}^{M})}$, $\mathcal{S}_{\text{ext},t} \leftarrow \{\mathbf{s}_{\text{ext},i}\}_{i \in \text{ANN-Search}(q_{\text{ext},t}, N; \mathcal{I}_{\text{ext}})}$.
10:     **Similarity-weighted aggregation:** $\alpha_{t,j} \leftarrow \text{softmax}_j(q_{\text{int},t}^\top \mathbf{k}_{\text{int},j} / \sqrt{d_{\text{key}}})$, $m_{\text{int},t} \leftarrow \sum_j \alpha_{t,j} \mathbf{s}_{\text{int},j}$; obtain $m_{\text{ext},t}$ analogously.
11:     **Bi-directional cross-memory refinement (AIM):** $m'_{\text{int},t}, m'_{\text{ext},t}$ via the two cross-attention updates and LayerNorm.
12:     **Projection and gated fusion:** $o_{\text{int},t} \leftarrow m'_{\text{int},t} W_{m,\text{int}}$, $o_{\text{ext},t} \leftarrow m'_{\text{ext},t} W_{m,\text{ext}}$; $g_{\text{int},t} \leftarrow \sigma([o_{\text{prelim},t}; o_{\text{int},t}] W_{g,\text{int}})$, $g_{\text{ext},t} \leftarrow \sigma([o_{\text{prelim},t}; o_{\text{ext},t}] W_{g,\text{ext}})$; $o_{\text{final},t} \leftarrow o_{\text{prelim},t} + g_{\text{int},t} \odot o_{\text{int},t} + g_{\text{ext},t} \odot o_{\text{ext},t}$.
13:     **Predict & recurrent update:** $\mathbf{L}_t \leftarrow o_{\text{final},t} W_{\text{vocab}}$; $y_t \leftarrow \text{Decode}(\mathbf{L}_t)$; $u_{t+1} \leftarrow \text{Embed}(y_t)$; update SSM state to $x_t$ using $(x_{t-1}, u_t)$.
14:     **(Optional) Online consolidation:** periodically buffer recent decoder states into $\mathcal{M}_{\text{int}}$.
15: **end for**

---

## D   Language Modeling Experimental Details

**Dataset Details**   We use the following datasets and splits for our finetuning experiments:

- **PG-19** (Rae et al., 2019): We use the official, standard splits for this dataset. The models are finetuned on the official training set and evaluated on the validation and test sets.

- **ProofPile** (Azerbayev et al., 2023): As the full dataset is very large, we created smaller, representative splits for finetuning. We randomly selected 10,000 samples from the official training set to create our finetuning set. For validation and testing, we randomly sampled 1,000 samples from the official validation set and 1,000 samples from the official test set, respectively.

- **CodeParrot** (Thomas Wolf & Zebaze, 2023): The official CodeParrot dataset does not provide predefined training, validation, or test splits. To create a consistent benchmark, we randomly sampled 100,000 samples for our finetuning training set, 1,000 samples for our validation set, and 2,000 samples for our test set.

**Training Setup**   We finetune each model on a total of 100M tokens. We use a constant learning rate of 1e-4, a global batch size of 128 (using batch accumulation), and the AdamW optimizer with a weight decay of 0.1 and gradient clipping of 1.0. During training, we sample a single window with a context length of 4k tokens from each example. During evaluation, for each example, we evaluate 10 windows with a maximal constant stride. We measure perplexity on only the last 100 tokens in each window to specifically test the model's extrapolation abilities.

**External Knowledge Base**   For **PG-19**, we constructed the external memory from an English Wikipedia snapshot (Wikimedia Foundation, 2012), which provides background knowledge about historical events, literary works, and cultural references beyond the book corpus. For **ProofPile**, we assembled an auxiliary mathematical reference corpus consisting of arXiv mathematics papers

(Ginsparg, 2001) and Wikipedia mathematics pages (Wikimedia Foundation, 2012), enabling access to formal definitions, theorems, and proofs not explicitly contained in the training set. For **Code-Parrot**, we extracted Python-specific files from *The Stack* dataset (Kocetkov et al., 2022), thereby incorporating a large-scale source of open-source Python code to provide knowledge of libraries and idiomatic coding patterns beyond the CodeParrot training data.

### D.1 Implementation Details for Comparative Models

To ensure a fair and direct comparison for the perplexity benchmarks in Table 1, we standardized the experimental setup for all model variants.

- **Parameter Count**: All models, including the Transformer$_{\text{full\_attn}}$ baseline, were configured to have approximately 130 million parameters.
- **Mamba Variant Construction**: The attention-augmented Mamba models and our PUM-Net were constructed upon the same Mamba-130M backbone. To create each variant, we randomly selected 20 of the original Mamba blocks and replaced them with the corresponding new architectural block. For example:
  - To build the **Mamba w/ SWA** model, 20 Mamba blocks were substituted with 20 Sliding Window Attention (SWA) blocks.
  - To build the **Mamba w/ NSA** model, 20 Mamba blocks were substituted with 20 Native Sparse Attention (NSA) blocks.
  - To build our **PUM-Net** models, 20 Mamba blocks were substituted with 20 of our proposed PUM-Net blocks.

  This block-replacement strategy, while representing a substantial architectural modification, is designed to fairly compare the efficacy of different block types (SWA, NSA, PUM-Net) within the same 130M-parameter framework, ensuring that the primary variable under investigation is the block architecture itself.

## E Passkey Retrieval Experimental Details

**Task Setup**   To specifically isolate and evaluate the model's long-context recall capabilities, we use the passkey retrieval task, a synthetic "needle-in-a-haystack" benchmark. Our setup follows the methodology described in Ben-Kish et al. (2024). The task requires the model to retrieve a 5-digit code embedded at a random sequence depth within a long document. The distractor text for these documents is sourced from samples in the WikiText dataset (Merity et al., 2016). Since the answer is always present in the input, success on this task depends solely on the model's ability to access and recall information from its context, not on external knowledge. Given that this task exclusively tests in-sequence recall, we use our `PUM-Net (w/o ex)` variant for a direct comparison against a suite of strong baseline models. These baselines include the original Mamba (Gu & Dao, 2023), as well as its variants DeciMamba (Ben-Kish et al., 2024) and MambaExtend (Azizi et al., 2025). We evaluate all models at two different scales: 130M and 1.4B parameters. To establish a strong baseline for comparison, all models were fine-tuned for one epoch on a dataset with a 4k context length. All other experimental parameters are consistent with the Training Setup detailed in Appendix D.

**Evaluation Protocol**   The evaluation is conducted across a wide range of sequence lengths and passkey depths to thoroughly probe the models' performance. We test context lengths of 1K, 2K, 4K, 8K, 16K, 32K, and 64K tokens. For each context length, the passkey is hidden at relative depths of 0%, 25%, 50%, 75%, and 100% of the sequence.

A retrieval is considered successful if the model generates the passkey verbatim. We compute the overall *retrieval score* for each model by assigning a score of 1 for each correct retrieval and 0 for each incorrect one, and then averaging across all tested depths and context lengths. The score is presented as a percentage using the formula:

$$\text{Retrieval Score (\%)} = \frac{\text{Total correct retrievals}}{\text{Total (correct + incorrect) retrievals}} \times 100$$

## F  LONGBENCH QA EXPERIMENTAL DETAILS

**Benchmark and Finetuning**   The **LongBench** benchmark (Bai et al., 2023) contains a diverse suite of tasks designed to evaluate long-context understanding. Our experiments focus on the question-answering categories listed in Table 2. All models originate from the same instruction-tuned checkpoint, `xiuyul/mamba-2.8b-zephyr`. The **Mamba-2.8B** baseline is a direct fine-tuning of this base model. Our **PUM-Net (w/o ex)** and **full PUM-Net** variants were created by modifying this Mamba-2.8B architecture, replacing each of its original Mamba blocks with our proposed PUM-Net blocks. Subsequently, all three models (the baseline Mamba, and the two PUM-Net variants) were individually fine-tuned on the official training set for each respective LongBench task, using a fixed context length of 4k tokens. All other experimental parameters are consistent with the Training Setup detailed in Appendix D.

**External Memory Construction**   To provide the model with highly relevant external knowledge for each reasoning task, we constructed a targeted corpus for the external memory. Specifically, for each question within the LongBench QA tasks, we used the question as a search query against an index of the English Wikipedia. We then took the top-1 most relevant Wikipedia page from the search results. The full text content of this page was then encoded and stored, serving as the dedicated external knowledge source for the `PUM-Net (full)` model when answering that specific question.

**Evaluation Metrics and Results Interpretation**   Performance reported in Table 2 is measured using the F1 score, following the official LongBench protocol. To ensure clarity, we detail the metrics as follows:

- The columns '0-4k', '4-8k', and '8k+' report the **macro-averaged F1 score** for all test samples that fall within those respective context length groups. This provides a granular view of performance as context grows.
- The 'LB' column represents the official overall score for each task, serving as the primary benchmark metric.
- A value of 'N/A' is used for tasks (e.g., `Musique`) where the official benchmark does not provide a breakdown of scores by context length, although an overall 'LB' score is still computed.

While these results are based on a single training run for each model, the performance gains from PUM-Net are not isolated to a single task or context length. The improvements are observed **consistently and substantially** across a diverse suite of QA tasks (Single-Doc, Multi-Doc, and Few-Shot). This consistency, combined with the large margins of improvement in many cases, provides strong evidence for the robustness and effectiveness of our proposed architecture.

## G  EFFICIENCY BENCHMARK DETAILS

**Benchmark Setup**   To provide a fair and direct comparison of computational costs, the efficiency benchmarks shown in Figure 5 were conducted on a single, representative block of each architecture. We compare our **PUM-Net** block against a **Native Sparse Attention (NSA)** block (Yuan et al., 2025) and a **Flash-Attention** block (Dao et al., 2022). To ensure a fair comparison of architectural overhead, all benchmarked blocks were configured to a 130M parameter scale. All experiments were conducted on a single NVIDIA H100 GPU, measuring wall-clock time for forward/backward passes and peak allocated memory.

**Rationale for Baseline Selection**   Our choice of baselines was motivated by key conceptual similarities to our approach, allowing for a meaningful comparison of efficiency for long-context modeling:

- **Native Sparse Attention (NSA)**: The core idea in our internal memory mechanism—partitioning the sequence into chunks to process local information—was inspired

by the block-wise sparse patterns utilized in NSA. Therefore, comparing against it is crucial to show the efficiency gains of our SSM-based approach over a sparse-attention-based one.

- **Flash-Attention**: While it implements a mathematically equivalent dense attention, Flash-Attention's groundbreaking memory optimization is achieved by processing the computation in a block-wise or "tiled" fashion. We include it as a baseline as it represents the de facto standard for highly optimized Transformer implementations.

**Analysis of PUM-Net's Efficiency Advantage** PUM-Net's superior efficiency, especially at longer sequence lengths, stems from two core design principles of its dual-memory system:

1. **Linear Time Complexity**: The fundamental architecture of our PUM-Net block is derived from the Mamba model, which has linear time complexity ($O(L)$) with respect to sequence length $L$. This inherent efficiency means that its processing time and memory usage do not grow quadratically as sequence length increases, unlike attention-based mechanisms. This explains why the inference and training times scale far more favorably for PUM-Net.

2. **Zero-Overhead External Memory During Training**: Our external memory system is explicitly designed to avoid introducing computational overhead during the training loop. The knowledge base is pre-computed offline into a static set of encoded key-value pairs and indexed using a highly efficient Approximate Nearest Neighbor (ANN) library for fast lookups. Crucially, the retrieval process does not involve fetching raw text and prepending it to the input sequence—a common practice in retrieval-augmented models that would significantly slow down training. Instead, our retrieval mechanism is optimized directly into the model's parameters, allowing it to leverage external knowledge with no substantial additional time cost per training step.

## H ON THE CHALLENGES OF COMPARING WITH RAG METHODS FOR LONG-CONTEXT INPUTS

We considered including a direct comparison to traditional Retrieval-Augmented Generation (RAG) methods, as PUM-Net's use of an external memory shares a conceptual goal with retrieval augmentation. However, we concluded that a direct comparison is not straightforward due to a fundamental mismatch in the problem formulation, particularly concerning the handling of long-context inputs, which is the primary focus of our work.

**The Challenge of Dense Retrieval with Long Queries.** Standard RAG pipelines are designed for short, focused queries. They typically employ a dense retriever (e.g., a BERT-based bi-encoder) to map a query to a vector and retrieve text chunks with high semantic similarity. This is effective for short queries that produce a distinct semantic representation, leading to clearly distinguishable similarity scores. However, this paradigm breaks down for a very long query (e.g., 64k tokens). The semantic representation of such a long query becomes diffuse and less focused. When compared against a relevant document in a corpus, a long query will often exhibit high semantic overlap with nearly *all* chunks from that document. This results in undifferentiated, high similarity scores across many chunks, causing a loss of the retriever's discriminative power and making it difficult, if not impossible, to select a small, targeted set of "top-k" relevant passages.

**The Prohibitive Cost of a Chunk-and-Retrieve Strategy.** An alternative strategy would be to partition the long query itself and perform retrieval for each query chunk. However, this approach is computationally prohibitive and counter-productive. For instance, dividing a 64k-token query into 64-token chunks would result in 1024 individual queries. If we retrieve just one top-1 64-token text snippet for each of these query chunks, we would accumulate an additional 64k tokens of retrieved text. Concatenating this to the original input would create a 128k-token sequence for the model to process. This approach not only doubles the sequence length—exacerbating the very problem long-context models aim to solve—but also makes the training and inference costs untenable.

**Incompatibility with Pre-training Methods like REALM.** This fundamental limitation also applies to pioneering methods that integrate retrieval into the training loop, such as REALM (Guu et al.,

2020). While REALM effectively demonstrates how retrieved knowledge can be used to jointly optimize a model's parameters, its mechanism still relies on concatenating retrieved documents to the original input for the forward and backward passes. This concatenation-based augmentation, while powerful for short inputs, is fundamentally challenging to scale to the very long sequence lengths explored in our work and would not be computationally feasible.

**Conclusion.** In summary, due to these inherent challenges in applying existing RAG paradigms to long-sequence inputs, we determined that a direct experimental comparison would not be meaningful or fair. PUM-Net is designed to address a different challenge: efficiently augmenting an *already long, contiguous context* with *pre-encoded* external knowledge, rather than augmenting a *short query* with retrieved text. Therefore, we focused our comparisons on other state-of-the-art long-context architectures.

