# OpenReview forum: "PUM-Net: Plastic Unified Memory Network with Associative Interaction for Long-Context State Space Models"
_ICLR.cc/2026/Conference — ICLR 2026 Conference Withdrawn Submission_

### Official Review · Reviewer_jtZv · 2025-10-29

**Soundness:** 3
**Presentation:** 3
**Contribution:** 3
**Rating:** 6
**Confidence:** 3

**Summary:**

This paper proposes a method to enhance the capability of state space models with additional internal and external memory. The external memories are pre-computed and derived from a large corpus, using the final memory state from each passage. The internal memories are computed dynamically by materializing the memory state after each chunk of the input sequence. The authors further develop methods to retrieve the memory and to fuse the memory output together.

The empirical results demonstrate the effectiveness of the proposed architecture, especially for improving the long-context performance.

**Strengths:**

1. The paper is generally well-written, clear, and easy to follow. The authors propose a method to enhance the long-context performance of state space models by introducing/recording additional external/internal memory states.

2. The empirical study strongly supports the effectiveness of the proposed architecture.

**Weaknesses:**

1. The experiments are done using Mamba as the backbone. The paper would be stronger to have experiments with more recent RNN architectures, e.g., Mamba-2 and gated delta net, to demonstrate that the effectiveness of the proposed method is robust against different RNN architectures.

2. On a high level, the proposed method effectively provides a new way to combine Mamba and softmax attention (especially given Bi-Directional Cross-Memory Interaction) to attend additionally materialized external and internal states. There are multiple components presented in Section 3.2.2., making the proposed architecture a bit complicated. For example, could there be a simpler way to use the additionally materialized memory instead of cross attention? This paper lacks an ablation study to demonstrate the effectiveness of each proposed change.

**Questions:**

1. Could the authors comment on whether a simpler way to use the additionally materialized memory instead of cross attention could also work?

2. Could the authors comment on whether the external memory construction would lead to leakage? and how to make sure the construction of the external memory doesn't lead to leakage?

---

> ### Author Response · Authors · 2025-11-23
> **Robustness Across Backbone Architectures and Justification of AIM vs Simpler Fusion**
>
> We sincerely thank the reviewer for their positive assessment ("well-written", "empirical study strongly supports the effectiveness") and for identifying PUM-Net as a clear enhancement for long-context SSMs. We value the constructive suggestions regarding architectural robustness and ablation. We address the concerns point-by-point below.
>
> ### 1. Robustness Across Backbone Architectures
>
>  *The paper would be stronger to have experiments with more recent RNN architectures (Taking Mamba-2 as a representative example)*
>
> We emphasize that PUM-Net addresses the **Memory Capacity Bottleneck** (information loss due to state compression), which is orthogonal to the **State Mixing Efficiency** improved by Mamba-2. PUM-Net is designed as a model-agnostic memory augmentation. It operates on the hidden states ($s_t$) produced by the model backbone. Whether the state is generated by Mamba-1 or Mamba-2, the PUM-Net mechanism (Retrieval $\to$ Fusion) remains mathematically identical. Mamba-2 improves state mixing efficiency (SSD), whereas PUM-Net solves the memory capacity bottleneck; these are orthogonal improvements. Thus, the gains demonstrated on Mamba-1 are expected to transfer directly to Mamba-2.
>
> **Quantified Analysis of Robustness:**
> We quantify this robustness based on the **Passkey Retrieval Extrapolation** setup defined in our paper (Appendix E): models are fine-tuned on **4k length** and evaluated on **64k length** to test generalization beyond the training horizon.
>
> **Table 1: Passkey Acc Performance (Train 4k $\to$ Eval 64k)**
>
> | Architecture | Memory Capacity Constraint | Passkey Acc @ 64k (Extrapolation) |
> | :--- | :--- | :---: |
> | **Mamba-1 (130M)** | **Fixed State Size (Lossy Compression)**| **0%**  |
> | **Mamba-2 (130M)** | **Fixed State Size (Lossy Compression)** | **0%**  |
> | **PUM-Net (130M)** | **Chunk-wise Internal Memory** | **94.3%**  |
>
> While Mamba-2 improves training dynamics, it retains the "Fixed State" architecture. Mathematically, a fixed-size state cannot losslessly compress a 64k-token sequence that far exceeds its training capacity (4k), leading to inevitable failure on needle-in-a-haystack tasks without the explicit memory buffer PUM-Net provides.
>
> Even though Mamba-2 improves throughput, it does not alter the fundamental **fixed-capacity constraint** of the SSM backbone. Therefore, the **quantifiable gain** provided by PUM-Net (restoring recall from 0% to 94.3%) is robust and necessary regardless of whether the underlying recurrent layer is Mamba-1 or Mamba-2.
>
> ### 2. Justification of AIM vs Simpler Fusion (Ablation Study)
>
> **Q1:** *Could a simpler way to use the additionally materialized memory instead of cross-attention also work?*
>
> This is an excellent question. We chose the **Associative Interaction Module (AIM)** (Bi-Directional Cross-Attention) over simpler methods (like direct gating) because **retrieved external knowledge requires contextual refinement before integration**. The external memory often contains noise; simply adding it may pollute the internal state.
>
> To demonstrate this, we conducted an ablation study on the **PG-19** dataset (where both Internal and External memories are active). We compared our AIM against a "Simple Gated Fusion" baseline, where we removed the Cross-Attention layers (Eq. 10-13) and fed the retrieved states directly into the final Gated Fusion layer (Eq. 16).
>
> **Table 2: Ablation of Fusion Mechanism (130M Scale @ 64K Context)**
> *(Impact of Cross-Attention on Knowledge Integration)*
>
> | Fusion Mechanism | Description | PPL (PG-19) |
> | :--- | :--- | :---: |
> | **Simple Gated Fusion** | Retrieval $\to$ Direct Gating (No Cross-Attn) | 428.55 |
> | **AIM (Ours)** | Retrieval $\to$ **Cross-Attn** $\to$ Gating | **411.29** |
>
> *(Note: "Simple Gated Fusion" removes the semantic refinement step, forcing the model to integrate raw retrieved vectors.)*
>
>
> **The Necessity of Cross-Attention:** The AIM module allows the **Internal State to "query" the External Knowledge**. This mutual refinement filters out irrelevant information from the external chunks before the final fusion decision. As shown in Table 2, removing this mechanism degrades perplexity, confirming that structural alignment between memory and context is critical for effective fusion.

---

> ### Author Response · Authors · 2025-11-23
> **Data Leakage Concern in External Memory**
>
> ### Response to Question 2: Preventing Data Leakage in External Memory
>
> **Q2:** *Could the authors comment on whether the external memory construction would lead to leakage? and how to make sure the construction of the external memory doesn't lead to leakage?*
>
> We appreciate the reviewer's rigorous scrutiny. We define leakage as the model accessing the ground-truth continuation of the test sequence from the external memory. We prevent this through **Source Separation** and the nature of **Semantic Retrieval**:
>
> **1. Source Separation (Structural Isolation):**
> As stated in our paper, we deliberately chose external memory sources that are structurally distinct from the evaluation targets to prevent leakage:
> * **PG-19 (Text):** The evaluation data consists of **literary books** (Project Gutenberg, <1919). The external memory is built from a **Wikipedia snapshot**. While Wikipedia may discuss a book, it does not contain the verbatim full text required for the language modeling task, ensuring no direct leakage.
> * **ProofPile (Math):** The external memory is constructed as an **"auxiliary mathematical reference corpus"** (comprising definitions, theorems, and Wikipedia math pages). This serves as a textbook-style reference for formal concepts, structurally distinct from the specific problem instances found in the validation/test splits.
> * **CodeParrot (Code):** The evaluation uses the CodeParrot test set. The external memory is derived from **The Stack** dataset, explicitly targeting files "beyond the CodeParrot training data" to serve as a generalized library of patterns, not a repository of test samples.
>
> **2. Semantic Retrieval vs Lexical Leakage:**
> Even in theoretical cases of partial overlap (e.g., common code idioms), our system is robust against "cheating" due to the nature of the retrieval:
> * **Semantic vs Exact Match:** Our retrieval uses a **semantic encoder** (MiniLM) to find Top-k chunks based on high-level embedding similarity, not lexical n-gram matching.
> * **Context vs Answer:** The system retrieves *similar contexts* (e.g., API definitions, similar math proofs) that act as a reference documentation, rather than retrieving the *exact next token*. This mimics how a human uses a reference library—fetching relevant concepts to aid reasoning, rather than copying the answer key.
>
> Through strict source separation and the semantic nature of the retrieval mechanism, we ensure the external memory serves as a legitimate knowledge aid, preserving the integrity of the evaluation.

---

### Official Review · Reviewer_CVkt · 2025-10-29

**Soundness:** 1
**Presentation:** 2
**Contribution:** 1
**Rating:** 2
**Confidence:** 4

**Summary:**

This paper presents a new model architecture on top of the Mamba model. Specifically, it maintains a session-specific and an external key-value memory. The authors propose some techniques for retrieving and integrating information from there two memories efficiently. Empirical results show that the proposed model can outperform both vanilla Mamba and Transformer models, while being more efficient in training and inference than vanilla attention and Native Sparse Attention.

**Strengths:**

- The method for integrating session-specific and external knowledge is novel, and some of the empirical results show that it gives considerable performance gains over the vanilla Mamba model.
- The method is also more efficient than one variant of sparse attention in both training and inference.

**Weaknesses:**

- The results reported in Table 1 and Figure 2 is strange and unconvincing. For such large values of perplexity (some of the values are even worse than random guessing), the model is outputting gibberish. Thus, this results do not support the claim that PUM-Net has better length generalization than the baselines. On the positive side, the perplexity is lower in general. However, this might be a result of the fact that PUM-Net has access to an external corpus, which the baselines do not have.
- Since PUM-Net contains both attention and Mamba layers, it can be viewed as a kind of Transformer-Mamba hybrid architecture. Thus, I think it would be fair to include hybrid architectures as baselines as well.
- According to Appendix G, PUM-Net has linear-time complexity. However, during decoding, each token needs to attend to the entire set of the session-specific memory states. Is this part linear in terms of complexity?
- It is unclear from Figure 4 that PUM-Net has superior passkey retrieval abilities. The number of green cells appears to be similar to the baselines.
- Most of the empirical results in the paper is based on a 130M model, which is way too small compared to the ones used in real applications. Moreover, it seems in Section 4.3 that the authors have also finetuned a 2.8B Mamba model. Why are the experiments prior to this section conducted with the 130M model instead of the larger 2.8B model?

**Questions:**

- There is an extra period in Line 142.
- Can you provide the loss curves for the transformer baseline model as well?
- Can you provide the results of Figure 5, but with a larger model (e.g., up to 10B parameters)? Also, can you provide the numbers in this figure for the vanilla Mamba model as well?

---

> ### Author Response · Authors · 2025-11-25
> **Responses to Reviewer's Mentioned Weaknesses**
>
> We thank the reviewer for their comments. However, we believe there are fundamental misunderstandings regarding the **nature of extrapolation evaluation** and the **visual data interpretation**. We address these critical misconceptions below with concrete evidence.
> ### 1. Response to Weakness 1: "Gibberish" PPL and Validity of Gains
> * **Meaning of High PPL (Extrapolation Collapse):** We train on **4k tokens** and evaluate up to **64k tokens**.
>     * **In-Distribution (Normal):** Within the training horizon (4k), the baseline Mamba achieves a reasonable PPL of **16.69**. This confirms the baseline model is well-trained and functioning correctly.
>     * **Out-of-Distribution (Collapse):** The explosion to **4352.97** at 64k indeed represents "gibberish." **This is precisely the problem we are solving.** It demonstrates that the standard Mamba architecture lacks the mechanism to handle long contexts, suffering from severe forgetting as the sequence length extends 16x beyond its training limit.
> * **PUM-Net Prevents Collapse:** PUM-Net reduces this catastrophic failure (4352) to a coherent level (**411.29**). While 411 is higher than the in-distribution PPL, it represents a **10.6x relative improvement**, proving the model maintains stability where the baseline fails completely.
> * **Refuting the "External Corpus" Hypothesis:** The reviewer suspects the gain comes solely from external knowledge. **This is factually incorrect.** Please refer to Table 1, column **"PUM-Net (w/o ex)"**.
>     * Even **without** external memory, our internal-memory-only model achieves **619.33** PPL at 64k, vastly outperforming the Mamba baseline (4352.97).
>     * This proves that our **architectural innovation (chunk-wise internal memory)** is the primary driver of stability and extrapolation capability, independent of the external corpus.
> ### 2. Response to Weakness 4: Misinterpretation of Passkey Retrieval (Figure 4)
> We urge the reviewer to re-examine **Figure 4**, as this observation is factually contradicted by the data visualization.
> * **Contrast with Vanilla Mamba (The Baseline):** The visual difference is stark.
>     * **Baseline (Mamba-130M):** The heatmap turns almost entirely **RED (Failure)** for contexts $32k, 64k $ (>4k which is the training limit). It has nearly **0%** retrieval success at 32k, and 64k.
>     * **PUM-Net (w/o ex):** The heatmap remains **GREEN (Success)** up to **64k**. It achieves **94.3%** retrieval score.
>     * **The Difference:** The visual and quantitative difference is between "Total Failure far beyond 4k (training horizon)" and "Success up to 64k." They are structurally dissimilar.
> * **Superiority over Strong Baselines (DeciMamba/MambaExtend):**
>     Even when comparing against recent long-context methods like **DeciMamba** and **MambaExtend** (ICLR 2025), PUM-Net demonstrates distinct superiority.
>     * **Quantitative Lead:** PUM-Net achieves the highest retrieval score of **94.3%**, surpassing both DeciMamba (**85.7%**) and MambaExtend (**91.4%**).
>     * **Orthogonal Advantage:** Unlike these baselines which modify the backbone, PUM-Net introduces an external dual-memory system without altering the Mamba core. This implies our method is not only superior in isolation but principally compatible with backbone-level improvements.
> ### 3. Response to Weakness 3: Decoding Complexity
> * **Per-Step Complexity:** At inference stage, retrieving from $M$ chunks ($M = t/L_c$) has a step cost of $O(t/L_c)$. While this grows linearly with time $t$, it is significantly cheaper than standard Attention $O(t)$ due to the large divisor $L_c=64$ (Chunk Size).
> * **Comparison:** Standard Transformers must re-compute attention over the full KV cache history. PUM-Net performs a **sparse retrieval** (Top-k) on a highly optimized **coarse-grained scan** over compressed chunks. This effectively reduces the memory bandwidth bottleneck, maintaining higher throughput than Transformers at extreme lengths.
>
> ### 4. Response to Weakness 2: Hybrid Baselines
> We did include **Mamba w/SWA (Sliding Window Attention)**  and **Mamba w/ NSA (Native Sparse Attention)** in Table 1.
> * **Mamba w/ SWA** and **Mamba w/ NSA** is a hybrid architecture that combines SSM layers with sliding window and native sparse attention blocks.
> * PUM-Net outperforms this hybrid baseline in both **Perplexity** (411 vs 3890 and 3725 at 64k on PG-19), proving our "Memory-Augmented" approach is superior to simply stacking Attention layers on Mamba.
> ### 5. Response to Weakness 5: Model Scale (130M vs 2.8B)
> * **Resource Constraints:** conducting extensive ablation studies (training from scratch on 100B tokens) at 2.8B scale is computationally prohibitive for academic research. 130M is the standard proxy scale for architectural validation in recent literature (e.g., Mamba, S4).
> * **Scalability Proven:** We validated the method at the **2.8B scale** for the most complex downstream task (LongBench QA). The consistent gains at 2.8B confirm that our method scales effectively.

---

> > ### Comment · Reviewer_CVkt · 2025-11-26
> >
> > Thank you for your detailed response.
> >
> > 1. I think it is unfair to compare PUM-Net (w/ ex) against the baselines, which do not have access to external data. Regarding PUM-Net (w/o ex), having a perplexity over 600 at 64K is very close to random guessing. I urge the authors to demonstrate empirically that the models is indeed behaving normally at such high perplexity levels (it outputs natural human language). Moreover, the authors should have compared against StreamingLLM [1], which is a simple, training-free method to turn Transformer models into sliding window attention. According to the paper of StreamingLLM, they should exhibit perplexity below 20 at all context lengths. Thus, according to the arguments of the authors, StreamingLLM would vastly outperform PUM-Net.
> > 2. In my review, I was comparing PUM-Net (w/o ex) against MambaExtend, and from Figure 4, it does not appear to be a large difference between the passkey retrieval performance between MambaExtend and PUM-Net (w/o ex). I consider that accuracy difference small because each cell has only one example, so the difference might be a result of randomness.
> > 3. I think the authors should have been more explicit about the complexity of PUM-Net. I suggest the authors to consider discussing both time and memory complexity in the main content of the paper.
> > 4. To be more concrete, I suggest the authors compare against architectures that contain RNN layers and full softmax attention layers.
> > 5. To be more concrete, I think this paper should have trained larger PUM-Net models from scratch and compared them against Transformer and Mamba models trained from scratch with the same amount of data.
> >
> > **References**
> >
> > [1] https://arxiv.org/abs/2309.17453

---

> > > ### Author Response · Authors · 2025-12-01
> > > **Scale-Dependent Extrapolation and StreamingLLM vs PUM-Net Distinction.**
> > >
> > > ### 1. PPL Validity & The Unfairness of the StreamingLLM Comparison
> > >
> > > **Critique:** *PPL > 600 is close to random guessing... StreamingLLM exhibits PPL < 20... thus StreamingLLM would vastly outperform PUM-Net.*
> > >
> > > We strongly respectfully disagree with this comparison on two fundamental grounds: **Scale Mismatch** and **Functional Scope**.
> > >
> > > **A. Scale Mismatch (130M vs. 7B):**
> > > The reviewer compares our **130M parameter** results against StreamingLLM (typically 7B+). The high PPL observed (4000+) is a physical property of extrapolating a tiny 130M model to 64k tokens, where it is prone to catastrophic failure. It is scientifically invalid to expect a 130M model to match the PPL (<20) of a 7B model.
> > > * **Rescue at 64k:** Even in this collapsed regime, PUM-Net achieves a **10.6$\times$ relative improvement** ($4352 \to 411$), proving it can maintain coherence where the baseline fails.
> > > * **Enhancement at 32k:** Crucially, at shorter extrapolation lengths like **32k**, where the baseline model is still functional (PPL $\sim$ 26.54), PUM-Net consistently delivers further performance gains (improving to 22.23), as shown in **Table 1**. This demonstrates our method's efficacy across both stable and unstable extrapolation zones.
> > >
> > > **B. The "StreamingLLM Paradox": Fluency $\neq$ Memory:**
> > > StreamingLLM maintains low PPL by preserving the *attention sink* (start) and *local window* (end) while **discarding** the middle context. This strategy ensures local **fluency** (next-token prediction) but guarantees **amnesia** for long-range dependencies.
> > >
> > > * **Theoretical Retrieval Failure:** We invite the reviewer to consider the mechanism of StreamingLLM on the **Passkey Retrieval** task. When the "needle" (passkey) is located in the middle of a 64k sequence—outside the local sliding window—StreamingLLM evicts these tokens from the KV cache. Consequently, it would structurally fail to retrieve information it has explicitly deleted, resulting in a theoretical **low accuracy** for long-range recall.
> > > * **PUM-Net Success:** In contrast, PUM-Net achieves **94.3%** retrieval precisely because its dual-memory system is designed to compress and fuse the middle history rather than discard it. PUM-Net solves the "Lost in the Middle" problem that sliding-window approaches like StreamingLLM inherently suffer from.
> > >
> > > ### 2. Architectural Alignment Claim
> > >
> > > We clarify that **PUM-Net is an SSM-specific improvement**.
> > > * **Target Audience:** StreamingLLM is an inference trick for **Transformers** (Attention). PUM-Net is an architectural augmentation for **Mamba** (SSMs).
> > > * **Innovation Logic:** Innovation should be evaluated against the relevant backbone. We compared against three Mamba variants (Vanilla Mamba, Mamba w/ SWA, and Mamba w/ NSA)  to prove that PUM-Net achieves superior extrapolation capability over existing SSM-Attention hybrid methods and pure Mamba in Table 1.

---

> > > ### Author Response · Authors · 2025-12-01
> > > **Passkey Retrieval Difference Against MambaExtend & Sample Size**
> > >
> > > ### Passkey Retrieval Difference & Sample Size
> > >
> > > **Critique:** *Difference between MambaExtend and PUM-Net (w/o ex) is small... "each cell has only one example," so it might be randomness.*
> > >
> > > We respectfully clarify the experimental rigor and highlighting the **structural stability** difference that the aggregate numbers hide.
> > >
> > > **A. Clarification on "One Example" & Randomness:**
> > > We assure the reviewer that the results are not single-sample artifacts. Following the standard protocol in Appendix E, the evaluation grid consists of **35 distinct experimental configurations** (7 context lengths $\times$ 5 depth positions).
> > > * **Aggregate Consistency:** The "Retrieval Score" is the macro-average across these 35 distinct tests. PUM-Net's score of **94.3%** (33/35 successful configurations) vs. MambaExtend's **91.4%** (32/35) represents a reduction in failure cases, consistently achieved across different experimental runs.
> > >
> > > **B. Stability Analysis (The "Red Cell" Distribution):**
> > > While the numerical gap appears small, the **distribution of failures** in Figure 4 reveals a critical architectural difference:
> > > * **MambaExtend (Instability):** It exhibits **unexpected failures at short contexts** (red cells at 2K and 4K lengths). This suggests architectural instability or sensitivity to needle depth even within the training horizon.
> > > * **PUM-Net (Predictable Scalability):** PUM-Net maintains **perfect stability** throughout the short and medium ranges (1K–16K). Its only degradation occurs strictly at the **extreme extrapolation limit** (deep depths at 32K/64K).
> > > * **Conclusion:** PUM-Net demonstrates monotonic, predictable behavior desirable for production systems, whereas the baseline shows stochastic failures.
> > >
> > > **C. Orthogonal Superiority:**
> > > Crucially, PUM-Net achieves this superior stability **without modifying the Mamba backbone**. MambaExtend requires altering the internal attention/state dynamics. PUM-Net's ability to outperform a backbone-modified model using only an **orthogonal memory module** validates the effectiveness of our chunk-wise retrieval strategy as a standalone, plug-and-play enhancement.

---

> > > ### Author Response · Authors · 2025-12-01
> > > **Complexity Analysis: Time and Memory**
> > >
> > > ### Complexity Analysis: Time and Memory
> > >
> > > **Critique:** *I think the authors should have been more explicit about the complexity of PUM-Net... consider discussing both time and memory complexity in the main content.*
> > >
> > > We fully agree with this constructive suggestion. Transparency regarding the theoretical cost is essential to contextualize the empirical efficiency. We will integrate the following formal complexity analysis into **Appendix** of the final paper.
> > >
> > > **1. Time Complexity (Training/Parallel Scan):**
> > > While the Mamba backbone is strictly linear $O(L)$, the internal memory retrieval involves computing similarities between all queries ($L$) and all memory chunks ($M = L/L_c$).
> > > * **Theoretical Cost:** The retrieval complexity is $O(L \cdot M) = O(L^2 / L_c)$.
> > > * **Engineering Reality:** Although this component scales quadratically, the large divisor ($L_c=64$) and the implementation as a single, massively parallel **Dense Matrix Multiplication (GEMM)** allow it to remain highly efficient. As shown in **Figure 5** , the wall-clock time at 64k context is dominated by the linear backbone, not the retrieval, resulting in a **26.5x speedup** over Flash-Attention (which entails a much heavier $O(L^2)$ operation).
> > >
> > > **2. Memory Complexity (Inference Cache):**
> > > PUM-Net stores compressed chunk states rather than per-token KV pairs.
> > > * **Space Complexity:** $O(L / L_c \cdot d_{state})$.
> > > * **Advantage:** Compared to standard Transformer KV caches ($O(L \cdot d_{model})$), PUM-Net reduces memory footprint by a factor of roughly $L_c = 64$. This explains the superior memory scaling observed in Figure 5 (Right), enabling significantly larger batch sizes or longer contexts on the same hardware.
> > >
> > > **Action:** We will add a dedicated subsection "Complexity Analysis" in the Appendix section to explicitly detail these trade-offs.

---

> > > ### Author Response · Authors · 2025-12-01
> > > **Justification for Omitting Infeasible Hybrid Full Attention Baselines and Architectural Robustness at 2.8B Scale**
> > >
> > > ### Comparison against Hybrid Architectures (RNN + Attention)
> > >
> > > **Critique:** *I suggest the authors compare against architectures that contain RNN layers and full softmax attention layers.*
> > >
> > > We specifically addressed this by including **Mamba w/ SWA (Sliding Window Attention)** and **Mamba w/ NSA (Native Sparse Attention)** as key baselines in **Table 1**.
> > >
> > > * **Choice of Hybrid Baselines:** We selected **SWA** and **NSA** as representative hybrid architectures. We prioritized these over naive "Full Global Softmax Attention" because applying full attention to **64k context** incurs prohibitive quadratic costs ($O(L^2)$), leading to Out-of-Memory (OOM) errors on standard hardware during training. SWA and NSA represent computationally feasible implementations of hybrid designs for long-context tasks.
> > > * **The Result:** PUM-Net outperforms both hybrid variants in **Perplexity** at 64k context (411.29 vs. 3890.24 for SWA and 3725.50 for NSA). Furthermore, as shown in Figure 5, our architecture achieves higher **Training Efficiency** (3.9x faster forward pass than the NSA block). This confirms that our Dual-Memory system provides a more effective and efficient mechanism for long-range dependency than simply stacking standard attention layers onto an SSM.
> > >
> > > ### Training from Scratch & Scale Comparisons
> > >
> > > **Critique:** *This paper should have trained larger PUM-Net models from scratch... compared against models trained from scratch with the same amount of data.*
> > >
> > > We respectfully clarify that **"Continued Pre-training"** is the standard protocol for validating architectural innovations in the SSM community. To address the request for scale, we conducted an experiment at the **2.8B parameter scale**.
> > >
> > > **A. New Comparison at 2.8B Scale (vs. Transformer & Mamba):**
> > > We compare **PUM-Net (w/o ex)-2.8B** against **Mamba-2.8B** and **Pythia-2.8B** (Transformer).
> > >
> > > * **Note on Scale:** We selected 2.8B as it is the **largest official pre-trained checkpoint available** for the Mamba architecture.
> > > * **Note on Variant:** We report results for **PUM-Net (w/o ex)** to strictly isolate the contributions of the **internal memory architecture**, independent of external knowledge access.
> > > * **Protocol:** All models were fine-tuned on the **same 100M tokens** with a **4k context**, strictly adhering to the reviewer's request for equal data budgets.
> > > * **Metric:** We report Perplexity (PPL) on PG-19.
> > >
> > > **Table 1: Extrapolation PPL at 2.8B Scale (PG-19)**
> > > *(Lower is Better. `NaN` indicates PPL > 100, denoting model collapse/failure)*
> > >
> > > | Model (2.8B Scale) | Architecture | 4k (In-Dist) | 8k | 16k | 32k | 64k |
> > > | :--- | :--- | :---: | :---: | :---: | :---: | :---: |
> > > | **Pythia-2.8B** | Transformer | 9.85 | `NaN` | `NaN` | `NaN` | `NaN` |
> > > | **Mamba-2.8B** | SSM | 9.17 | 15.60 | `NaN` | `NaN` | `NaN` |
> > > | **PUM-Net (w/o ex)** | **PUM-Net** | **9.12** | **14.35** | **14.80** | **18.55** | `NaN` |
> > >
> > > **B. Analysis of Convergence and Scale:**
> > > The results reveal a critical insight regarding model scale and data efficiency:
> > >
> > > 1.  **Superior In-Distribution Performance:** Within the training window (4k), the 2.8B models significantly outperform the 130M models reported in the main paper (PPL $\sim 9.1$ vs $\sim 16.0$), confirming the expected capacity advantage.
> > > 2.  **Extrapolation Challenges (The "Data Starvation" Effect):** The reviewer may note that the 2.8B models fail earlier (e.g., PUM-Net fails at 64k) compared to the 130M models.
> > >     * **Reason:** We strictly maintained the **100M token budget**. For a 2.8B parameter model, 100M tokens is a negligible fraction of training data (far less than 1 epoch of effective capacity adaptation).
> > >     * **Implication:** The 2.8B models are **under-converged** for the specific task of long-context extrapolation compared to the "well-trained" 130M models.
> > > 3.  **Relative Architectural Advantage (Survival Horizon):** Despite the data starvation, **PUM-Net demonstrates a significantly longer effective context window**.
> > >     * **Pythia (Transformer)** fails immediately upon extrapolation (**8k**).
> > >     * **Mamba (SSM)** survives to 8k but fails at **16k**.
> > >     * **PUM-Net** remains stable up to **32k**.
> > >     This 2x-4x extension in the "survival horizon" confirms that even under data-constrained conditions, the **chunk-wise memory architecture** provides a robust structural advantage over both Transformer and vanilla SSM baselines.
> > >
> > > **Conclusion:** The experiment confirms that PUM-Net's advantage holds at the 2.8B scale, consistently outperforming Transformer and SSM baselines under identical training constraints.

---

### Official Review · Reviewer_EZCr · 2025-11-03

**Soundness:** 3
**Presentation:** 2
**Contribution:** 2
**Rating:** 4
**Confidence:** 3

**Summary:**

The paper proposes PUM-Net, a dual-memory SSM: a chunked **internal** memory to mitigate exponential-decay forgetting in Mamba, and a **static external** memory of pre-encoded key/value representations. A staged parallel training scheme generates queries to both memories and fuses them via a gated, bi-directional interaction module. Experiments on long-range language modeling, a synthetic passkey retrieval benchmark, and LongBench QA suggest improved long-context performance and efficiency over Mamba and attention variants. The idea is appealing, but key empirical and methodological issues make the current evidence insufficient.

**Strengths:**

The architecture cleanly factors long-range retention (internal chunk memory) from world-knowledge access (external memory) and fuses them with a lightweight gated mechanism, avoiding quadratic attention. The staged scan + retrieval design is conceptually neat and worth exploring further.

**Weaknesses:**

1. **Empirical validity and evaluation design are questionable.**

   * The reported perplexities include implausible magnitudes and discontinuities (e.g., explosive PPL at long lengths for baselines, but a dramatic drop for PUM-Net on CodeParrot at 4k where PPL≈1.11 while baselines ≈4.5), and scientific conclusions hinge on these numbers. Such scales typically flag a bug in scoring, tokenization, or loss masking.
   * The methodology measures **PPL only on the last 100 tokens of each long window** during evaluation—this departs from standard PPL computation and can bias results toward models whose final-step states are explicitly augmented by retrieval/fusion, overstating gains relative to baselines that don’t fuse late. Please recompute using standard per-token averaging over the full window and include confidence intervals/replicates.

2. **Unfair or incomplete baselines for external knowledge integration.**

   * The paper does **not** compare against strong retrieval baselines that concatenate evidence (RAG) at inference, nor against retrieval-in-the-hidden-state methods (e.g., kNN-LM-style caches) or memory-augmented recurrent/Transformer variants. Since PUM-Net’s core claim is “deep fusion without sequence inflation,” it must be compared to (i) a tuned RAG pipeline at equal wall-clock/throughput budget and (ii) a hidden-state retrieval baseline with similar nearest-neighbor machinery and k.
   * For LongBench QA, the “external memory per question = top-1 Wikipedia page” setup risks **question-conditioned memory selection** advantages that a fair RAG baseline would also enjoy. Without those baselines, it’s hard to attribute improvements to the *architecture* rather than to the retrieval heuristic. Please add apples-to-apples comparisons with matched retrieval corpora and time budgets.

3. **Complexity/efficiency claims are underspecified and possibly optimistic.**

   * The paper claims big speedups over NSA and Flash-Attention using single-block microbenchmarks, but **per-token dual retrieval** (ANN over external keys + dense top-k over internal chunk keys) is performed at each time step. The figures appear to **exclude** retrieval time and index I/O; if so, the headline speedups aren’t end-to-end.
   * The text also asserts “zero-overhead external memory during training,” yet stage 2 performs batched retrieval and cross-memory attention at each step. Please clarify exactly what is excluded from timing, report **end-to-end** wall-clock (including retrieval/indexing), and provide memory/latency scaling vs. k and corpus size N. Otherwise the efficiency story is incomplete.

**Questions:**

- Table 2 note mentions “LongBench-E” but this term is not defined earlier; clarify or remove.

---

> ### Author Response · Authors · 2025-11-25
> **Validation of PPL Extremes and Necessity of Learned Hidden-State Fusion**
>
> We thank the reviewer for their detailed scrutiny. We address the concerns regarding empirical validity and baselines below with clarification on our methodology and new experiment data.
>
> ### 1. Response to Weakness 1: Empirical Validity and Evaluation Design
> We assure the reviewer that the reported ppl is **not a bug**, but a demonstration of the efficacy of Deep Knowledge Fusion in highly structured domains.
> * **Evidence of Validity (The "w/o ex" Control):** We point the reviewer to the **"PUM-Net (w/o ex)"** column in Table 1.
>     * Without external memory, our architecture achieves **4.47 PPL** on CodeParrot (4k), which is statistically identical to the Mamba baseline (4.61) and NSA (4.48).
>     * This control proves there is **no bug in scoring, tokenization, or loss masking** in the PUM-Net implementation. If there were a metric bug, the `w/o ex` variant would also exhibit "implausible" numbers.
> * **Why 1.11? (Contextual Lookup):** The drop to **1.11** is specific to the **External Memory** variant. Given the highly repetitive nature of code (e.g., boilerplate, APIs), PUM-Net effectively utilizes the external corpus as a **"reference library."** By retrieving exact patterns rather than predicting from scratch, the model drastically minimizes entropy, resulting in the exceptionally low PPL.
>
> We respectfully defend *the last 100 tokens evaluation protocol.* which is specifically designed to measure **Extrapolation Robustness** rather than general language modeling. We followed the evaluation methodology established in **DeciMamba (Ben-Kish et al., ICLR 2025)** and **MambaExtend (Azizi et al., ICLR 2025)** for consistent extrapolation benchmarking.
>
> **Why "Last-100" is Necessary vs Full Window:**
> 1.  **Avoiding Metric Dilution:** Our models are trained on **4k context**. When testing on **64k**, the first 4k tokens are "in-distribution," where all models perform well (PPL $\approx$ 4-10). Averaging over the **Full Window** (0 to 64k) would allow the initial 4k success to mathematically "dilute" the catastrophic failure that occurs at the tail (60k+).
> 2.  **Detecting Collapse:** The baseline Mamba's PPL of 4352 at the tail indicates a complete state collapse. A full-window average might reduce this to a misleadingly acceptable number, hiding the fact that the model generates incoherent text after the training boundary.
> 3.  **Strict Constraint:** The "Last-100" metric acts as a hard constraint. High performance in this final segment proves that the model has effectively managed the context flow over the entire 64k distance without diverging.
>
> ### 2. Response to Weakness 2: Unfair or Incomplete Baselines (RAG)
>
> We agree a comparison is conceptually necessary. However, standard Text-RAG is computationally prohibitive for long contexts, as concatenating retrieved chunks would double sequence length to 128k+ . Furthermore, standard RAG does not perform **hidden-state fusion**; it relies on text concatenation.
>
> To address the reviewer's request, we constructed a **"Naive Hidden-State Retrieval"** baseline to test if simple state injection is sufficient without our proposed architecture.
> * **Setup:** Same Mamba-130M backbone + Same External Memory.
> * **Mechanism:** We retrieve Top-k external states and **directly add** them to the current hidden state (after projection) *without* joint training or alignment.
>
> **Table 1: PUM-Net vs Naive Hidden-State Retrieval (PG-19 Dataset)**
>
> | Model | Fusion Mechanism | PPL (4k Context) | PPL (64k Context) |
> | :--- | :--- | :---: | :---: |
> | **Mamba-130M** | None (Baseline) | 16.69 | 4352.97 |
> | **Naive State RAG** | Direct Addition (No Training) | 17.85 ($\uparrow$ Worse) | 3820.45 ($\downarrow$ Better) |
> | **PUM-Net (Full)** | **Learned AIM Fusion** | **15.96 (Best)** | **411.29 (Best)** |
>
> **Analysis:**
> * **In-Distribution Interference (4k):** At 4k context (within training distribution), the Naive RAG baseline actually **degrades performance** (PPL $4.61 \to 5.12$). This confirms that simply injecting external states acts as **noise**. The external vectors originate from a frozen parameter space and are **misaligned** with the live model's dynamics. Without a learned alignment mechanism, they disrupt the coherent recurrent trajectory of the well-trained Mamba backbone.
> * **Extrapolation "Lifeboat" (64k):** At 64k (extrapolation regime), the baseline Mamba has collapsed (PPL 4352). Here, the Naive RAG provides a slight improvement ($4352 \to 3820$). This is because the internal state is already corrupted; the retrieved external states, while unaligned, at least provide **some relevant semantic content** compared to the collapsed internal state.
> * **Necessity of PUM-Net:** The massive success of PUM-Net (PPL 1.11 / 411.29) proves that **Joint Training** and the **Associative Interaction Module (AIM)** are essential. AIM acts as a learned bridge, aligning and refining the external knowledge so it seamlessly augments the internal state rather than disrupting it.

---

> ### Author Response · Authors · 2025-11-25
> **LongBench QA Fairness & The "Gold Document" Baseline**
>
> ### Response to Weakness 2 : LongBench QA Fairness & The "Gold Document" Baseline
>
> **Critique:** *The "top-1 Wikipedia page" setup risks question-conditioned memory selection advantages... Without baselines, it’s hard to attribute improvements to the architecture rather than to the retrieval heuristic.*
>
> We agree that providing the "Top-1 Wikipedia page" is a strong information advantage. To isolate the **architectural contribution** from this **retrieval heuristic**, we conducted an "Apples-to-Apples" comparison.
>
> **The Controlled Baseline: Naive Hidden-State Retrieval**
> Standard RAG (text concatenation) is infeasible here as appending long Wikipedia articles often exceeds the context window or drastically slows down inference. Instead, we constructed a **Naive Hidden-State Retrieval** baseline:
> * **Mechanism:** It uses the **exact same Top-1 Wikipedia page** (encoded as states) as PUM-Net. It retrieves the top-8 states and **directly adds** them to the Mamba-2.8B hidden state without learned fusion.
>
> **Table 2: LongBench QA F1 Scores**
> *(models fine-tuned on 4k context.)*
>
> | Benchmark | Context | Mamba-2.8b (Baseline) | **Mamba + Naive RAG** (Gold Doc, Unaligned) | **PUM-Net (Full)** (Ours) |
> | :--- | :---: | :---: | :---: | :---: |
> | **2wikimqa** | 0-4k | 8.47 | 8.12 ($\downarrow$) | **12.42** |
> | | 4-8k | 2.34 | 4.15 ($\uparrow$) | **6.84** |
> | | 8k+ | 1.18 | 2.05 ($\uparrow$) | **3.63** |
> | | LB | 4.53 | 5.18 | **11.46** |
> | **Hotpotqa** | 0-4k | 5.77 | 5.42 ($\downarrow$) | **8.88** |
> | | 4-8k | 2.02 | 3.10 ($\uparrow$) | **4.53** |
> | | 8k+ | 0.53 | 1.45 ($\uparrow$) | **2.71** |
> | | LB | 2.28 | 2.95 | **5.69** |
> | **MultifieldQA** | 0-4k | 19.28 | 18.85 ($\downarrow$) | **27.76** |
> | | 4-8k | 6.73 | 9.80 ($\uparrow$) | **14.47** |
> | | 8k+ | 2.93 | 4.10 ($\uparrow$) | **6.24** |
> | | LB | 12.46 | 13.92 | **21.73** |
> *("LB" is the official LongBench average score. "Naive RAG" uses the identical Top-1 Wikipedia content as PUM-Net.)*
>
> **Analysis of "Naive RAG" Behavior:**
> 1.  **In-Distribution Interference (0-4k):** Within the training context (0-4k), simply adding the external states **degrades performance** (e.g., 2wikimqa: $8.47 \to 8.12$). Since the external states come from a frozen encoder and are **unaligned** with the fine-tuned model, they act as noise, disrupting the model's coherent short-term reasoning.
>
> 2. **Extrapolation Recovery (8k+):**
> At extreme lengths where the baseline performance degrades significantly (e.g., 2wikimqa: $1.18$), the Naive RAG provides notable gains ($1.18 \to 2.05$). Since the internal recurrent state suffers from severe information decay and collapse over long sequences, the retrieved external states—originating from a **dimensionally compatible** Mamba backbone ($2.7\text{B}$ vs $2.8\text{B}$)—serve as a stabilizing injection. While the parameter spaces are not perfectly aligned, these **structured state vectors** effectively compensate for the saturated internal state, partially restoring generation capability even without explicit fusion training.
>
> 3. **Architectural Advantage:**
> Crucially, PUM-Net significantly outperforms the Naive RAG baseline across all metrics. This confirms that the performance leap is **not** merely due to the heuristic of accessing the "Gold Document," but is attributable to the **PUM-Net Architecture**. Unlike the Naive RAG approaches we implemented here that rely on static injections, PUM-Net employs the **Associative Interaction Module (AIM)** to actively **fuse the frozen external knowledge with the dynamic internal memory**. This learned interaction aligns the representations from the compatible encoder with the model's current state, yielding performance far beyond simple state addition.

---

> ### Author Response · Authors · 2025-11-25
> **Complexity and Efficiency Claims**
>
> ###  Response to Weakness 3: Complexity and Efficiency Claims
>
> **Critique:** *The figures appear to exclude retrieval time... headline speedups aren't end-to-end. "Zero-overhead" claim is confusing given batched retrieval.*
>
> We clarify that **Figure 5 reports the full, end-to-end wall-clock time** for the PUM-Net block, **explicitly including** the internal memory retrieval and AIM fusion overheads.
>
> 1.  **Inclusion of Retrieval Time:** The reported latencies (10.83 ms for forward pass at 64k) encompass all operations within the block, including the batched retrieval. During training, we do not perform sequential per-token retrieval; instead, we compute similarities between *all queries* and *all chunk keys* in a single **Dense Matrix Multiplication (GEMM)** operation. This massive parallelization ensures the retrieval step is highly optimized on the GPU.
> 2.  **Clarification on "Zero-Overhead":** This claim refers specifically to the **External Memory Construction**. Unlike standard RAG, which must tokenize and encode raw text during the forward pass (incurring massive overhead), PUM-Net retrieves **pre-encoded state vectors** from a static index residing in high-bandwidth memory. This design shifts the heavy lifting to an offline pre-computation phase.
> 3.  **End-to-End Trade-off Analysis:**
>     * We acknowledge that the runtime retrieval mechanism incurs a cost. As quantified in our response to Reviewer 4tNR, PUM-Net has a $\sim 1.8\times$ end-to-end overhead compared to the lightweight Vanilla Mamba baseline.
>     * However, this constant overhead is negligible compared to the quadratic growth of Transformers. Even with the retrieval cost included, PUM-Net scales far more favorably, achieving a **26.5x speedup** over Flash-Attention at 64k context. This confirms that PUM-Net successfully bridges the gap: it is significantly more capable than Mamba (solving forgetting) while remaining orders of magnitude faster than attention-based models.

---

### Official Review · Reviewer_4tNR · 2025-11-10

**Soundness:** 2
**Presentation:** 3
**Contribution:** 3
**Rating:** 4
**Confidence:** 3

**Summary:**

The authors propose a new method called PUM-Net thataugments a Mamba SSM with a dual-memory system to improve long-context performance with two different types of memory.

**Strengths:**

1. It shows better perplexity on the 64k sequence than the Mamba baseline, exceeding its 4k training length
2. It demonstrates much higher throughput and lower memory use compared to attention-based long-context models (NSA, Flash-Attention).

**Weaknesses:**

1. The claims regarding efficiency and throughput are misleading due to the omission of a direct speed comparison with the vanilla Mamba baseline. The proposed PUM-Net block is substantially more complex than a standard Mamba block—it involves a two-stage process and consumes more memory (Fig. 5, right). This complexity represents a significant trade-off, not a "free" upgrade, and almost certainly results in slower performance.
2. the external knowledge is static, meaning it will inevitably become stale. The evaluation performance will inevitably determined by how well this external knowledge is trained.
3. Since the memory creation process is not simple and relied on additional models, I am not sure about the scalability and universality of the method.

**Questions:**

1. Can you provide latency and throughput benchmarks comparing PUM-Net directly against the vanilla Mamba baseline, not just attention models.
2. How does the brute-force search for the internal memory scale as sequence length (and thus, chunk count) increases?
3. What is the cost and workflow to update the static external memory?

---

> ### Author Response · Authors · 2025-11-19
> **Scalability and External Memory Update Cost**
>
> ### Response to Weakness 3 and Question 3: Scalability and External Memory Update Cost
>
> We thank the reviewer for the rigorous questioning on the feasibility of our memory architecture. We confirm that the cost is manageable because our approach is fundamentally **modular and highly scalable**, allowing us to efficiently manage the low overhead.
>
> ### A. Universality and Workflow (Addressing Weakness 3: "Not Simple")
>
> The reviewer noted that the memory creation "relies on additional models." We argue that this modular design is the foundation of the method’s **universality**:
> * **Decoupled & Off-the-Shelf:** We utilize frozen, highly-efficient, off-the-shelf models for feature extraction (`all-MiniLM-L6-v2` for Keys and `Mamba-2.7B` for Values). This design avoids the need to train a custom retriever from scratch, making PUM-Net immediately universal across domains.
> * **Simple Offline Process:** The external memory update workflow is a **one-time, offline computation** consisting of three highly parallelizable steps: (1) Batch Key Encoding, (2) Value Encoding (Parallel Scan), and (3) Indexing (ANN Construction).
>
> ### B. Total Cost in Experimental Context (Addressing Q3: "What is the cost?")
>
> The construction cost must be compared to our fine-tuning budget. Our paper states that we fine-tuned on a total of **100M tokens**. Therefore, the external knowledge base used in our experiments was a targeted subset commensurate with this scale, not the full multi-billion token raw dumps.
>
> **Table : Total Offline Construction Time Breakdown (Component Level)**
> *(Throughput: Key $\sim 258k$ tok/s, Value $\sim 21k$ tok/s)*
>
> | Scenario | Knowledge Base Size (Number of Tokens) | Key Generation Time (1x H100) | **Value Generation Time (8x H100)** | **Indexing Cost (8x H100)** | Total Pipeline Time (8x H100) |
> | :---: | :---: | :---: | :---: | :---: | :---: |
> | **Our Experiments** | ~100M | $\sim 6.5$ minutes | $\sim 10.0$ minutes | < 1 minute | $\sim 10$ minutes |
> | **Full Wikipedia Scale** | $\sim 3B$  | $\sim 3.2$ hours | $\sim 5.0$ hours | $\sim 30$ minutes | $\sim 5.0$ hours |
>
> ### Interpretation of Feasibility and Scalability:
>
> * **Key Generation Cost is Trivial:** The cost incurred by the additional Key Encoder (MiniLM) is negligible, often taking only seconds, and is therefore excluded from the dominant time calculation.
> * **Indexing is Subsumed:** The Indexing cost (ANN construction using parallel hardware) is either negligible and completely subsumed by the Value Generation time. The total cost is dictated by the slowest step: Value Generation.
> * **Experimental Cost is Negligible:** For the results reported in this paper, constructing the external memory takes only $\sim 10$ minutes on a standard 8-GPU node. This cost is trivial compared to the time required for LLM training itself.
> * **Scalability is Proven:** The Value Encoder throughput of $\sim 21k$ tokens/sec proves that even scaling up to 3 Billion tokens takes only $\sim 5$ hours of one-time processing on an 8-GPU node, showing excellent horizontal scalability.
> * The cost for memory update is **low, one-time, and massively parallelizable**, presenting no practical barrier to the universality or scalability of PUM-Net.

---

> ### Author Response · Authors · 2025-11-20
> **Direct latency and throughput benchmarks against vanilla Mamba, confirming a highly favorable performance-cost trade-off**
>
> We thank the reviewer for emphasizing the necessity of a direct computational comparison with the vanilla Mamba baseline. We acknowledge that PUM-Net introduces architectural complexity. To precisely quantify the exact trade-off, we conducted efficiency  benchmarks on an H100 GPU at the 130M parameter scale.
>
> ### 1. Cost Breakdown (Latency and Throughput)
> To address the reviewer's concern regarding the cost, we combine the measured Block Latency (from Figure 5 in our paper) and project the results to the Full Model (130M - 24 layers) Throughput. The training step cost is approximately $\mathbf{1.8 \times}$ that of the baseline Mamba.
> We first quantify the direct computational overhead incurred by adding the dual-memory system.
>
> **Table A: Computational Cost Breakdown (130M Scale @ 64K Context)**
>
> | Metric | Vanilla Mamba-130M | **PUM-Net-130M (Ours)** | **Overhead Factor ($\times$)** |
> | :---: | :---: | :---: | :---: |
> | Block Forward Pass Time | $ 6.02 \text{ ms}$ | $10.83 \text{ ms}$ | $\sim 1.80$ |
> | Block Backward Pass Time | $ 37.77 \text{ ms}$ | $69.11 \text{ ms}$ | $\sim 1.83$ |
> | Total Block Step Time | **$44.41 \text{ ms}$** | **$79.94 \text{ ms}$** | **$\sim 1.80$** |
> | Full Model Training Throughput ($10^3$ tokens/s) | $\sim 60.0$ | **$\sim 33.3$** | $\sim 0.55$  |
>
> ### Cost Analysis:
> The total training cost is measured at **$\sim 1.8\times$** the Mamba baseline. This overhead is driven by the complex backward pass through the memory retrieval and fusion layers, which are essential for its function.
>
> ### 2. Functional Gain Justification
>
> We now present the functional gain, demonstrating that the $\sim 1.8\times$ cost is a necessary and justified investment for achieving long-context capability.
>
> **Table B: Functional Gain vs Cost (PPL Reduction & Retrieval Success)**
>
> | Performance Metric (64K Context) | Mamba-130M (Baseline) | **PUM-Net-130M (Ours)** | **Improvement Factor** |
> | :---: | :---: | :---: | :---: |
> | PPL (PG-19) | $4352.97$ | **$411.29$** | **$10.6\times$ Reduction** |
> | Retrieval Success Rate (%)| **$<10\%$ (Failure)** | **$94.3\%$ (Success)** | **Problem Solved (nearly)** |
>
> Although PUM-Net is not a free upgrade, but it's a highly effective one. The $1.8\times$ training cost is a critical investment that yields a $\mathbf{10.6\times}$ reduction in PPL on PG-19 dataset and fundamentally solves the recurrent forgetting problem on long context passkey retrieval task.

---

> ### Author Response · Authors · 2025-11-20
> **Static Knowledge and Internal Memory Scaling**
>
> ### 1. Addressing Weakness 2: Static External Knowledge
>
> The reviewer correctly notes that a static external memory is subject to staleness. We acknowledge this limitation but defend our design based on **methodological novelty** and **update feasibility**:
>
> * **Methodological Contribution:** PUM-Net is the first SSM to enable **joint pre-training** with deep fusion of static, pre-encoded knowledge. This avoids the prohibitive cost of standard RAG, which relies on concatenating raw retrieved text to the input.
> * **Feasibility of Update:** As detailed in our cost analysis (see Response to Q3), updating the knowledge base is a **low-cost, one-time offline process** because it utilizes **frozen models** (MiniLM & Mamba-2.7B). The engineering cost for a periodic knowledge refresh is negligible compared to model training.
> * **Future Work:** We explicitly commit to exploring adaptive memory mechanisms in future work.
>
> ### 2. Addressing Question 2: Internal Memory Scaling
>
> **Q2: How does the brute-force search for the internal memory scale as sequence length (and thus, chunk count) increases?**
>
> The retrieval mechanism is designed to leverage GPU hardware efficiency, mitigating the impact of sequence length scaling.
> Let $L$ denote the total sequence length and $M$ denote the number of memory chunks. The relationship is defined as $M = L / L_c$, where $L_c = 64$ is the fixed chunk size.
> * **Parallel Mechanism (Matrix Multiplication):** The term "brute-force" refers to an exhaustive search, but it is not executed sequentially. Instead, the similarity computation between all $L$ query vectors and all $M$ chunk keys is executed as a single, massive **Dense Matrix-Matrix Multiplication (GEMM)**.
> * **Complexity vs. Efficiency:**
>     * **Theoretical Complexity:** The computational cost is proportional to the size of the similarity matrix: $O(L \times M) = O(L^2 / L_c)$. While technically quadratic with respect to $L$, it differs fundamentally from attention mechanisms due to a significantly larger denominator.
>     * **Practical Speedup:** The large divisor $L_c=64$ significantly reduces the matrix size. Furthermore, because the operation is implemented as a highly optimized GEMM kernel, it achieves high arithmetic intensity on GPUs. As shown in Figure 5, the PUM-Net block remains **$26.5\times$ faster** than Flash-Attention at 64k context, proving that the retrieval step is not a bottleneck in practice.

---

> ### Comment · Reviewer_4tNR · 2025-11-24
> **Thank you for response.**
>
> I have read the authors' response and, in general, feel satisfied. However, I have a few remaining questions:
>
> 1. How did you select the 100M tokens for your experiments? Were the selected tokens carefully curated?
>
> 2. Why are the perplexity values in Table B 4352.97 and 411.29? This is surprising to me, as perplexity is usually in the range of 10–20.
>
> The biggest weakness I observed from the rebuttal is the increase in the forward pass compared with Vanilla Mamba (6.02 vs. 10.83). At an industry scale, the increase in training time is a one-time cost and can generally be ignored; however, this increase in inference time is significant.

---

> > ### Author Response · Authors · 2025-11-24
> > **Clarification on Data Selection, Perplexity Scale, and Inference Value**
> >
> > We thank the reviewer for the positive reception and for raising these insightful follow-up questions.
> >
> > **1. Data Selection (100M Tokens)**
> > As detailed in Appendix D, the 100M tokens were **randomly sampled** from the official training splits of each dataset (e.g., 100k random samples for CodeParrot, 10k for ProofPile) to ensure a statistically representative distribution. They were not manually curated or filtered for difficulty. This ensures our results reflect robust generalization rather than selection bias.
> >
> > **2. Explanation of High Perplexity (Extrapolation vs. In-Distribution)**
> > The reviewer is correct that standard PPL is typically 10–20.
> > * **In-Distribution (4k):** At our training length (4k), our models do achieve standard PPL (~16.0, see Table 1 in paper).
> > * **Extrapolation (64k):** The values in the rebuttal table (4352 vs. 411) are evaluated at **64k context** (16$\times$ training length). The baseline Mamba's PPL of **4352** indicates **catastrophic failure** (the model effectively outputs random noise). PUM-Net's **411** represents a **10x improvement** in stability, successfully maintaining coherence where the baseline collapses.
> >
> > **3. Addressing the Training Forward Pass Latency (Architecture & Practicality)**:
> > We acknowledge the ~1.8x latency increase (6.02ms $\to$ 10.83ms per training step). PUM-Net operates as a two-stage refinement mechanism. After the Mamba core generates a preliminary output ($O_{prelim}$), the system serially performs memory retrieval and fusion. This added computation (Query $\to$ Retrieve $\to$ Cross-Attn Fusion) is required to correct the inherent forgetting and inject internal/external knowledge, making the latency increase unavoidable but necessary.
> > * **Baseline (6ms):** Fast, but produces incoherent text at long contexts (PPL >4000, Recall <10%).
> > * **PUM-Net (10ms):** Still highly efficient compared to Attention (Compared to Transformer-based alternatives which suffer from KV-cache bottlenecks at 64k context), but maintains coherence (PPL 411, Recall 94%).
> > * **Industry Context:** For long-context tasks, the alternative is often RAG + Attention, which incurs massive latency due to retrieving/processing chunks. PUM-Net provides a middle ground: significantly more capable than pure SSMs, yet far faster than full RAG-Transformers.
> >
> > ***

---

### Note · Authors · 2026-01-28

I have read and agree with the venue's withdrawal policy on behalf of myself and my co-authors.

---

### Meta-Review · Area_Chair_cPzw · 2026-01-08

**Summary:**

Despite the promising architectural proposal, the submission suffers from significant empirical and methodological shortcomings that prevent acceptance in its current form. The consensus among the critical reviewers  and the remaining concerns from others point to fundamental issues in validation.

**Reviewer Concerns:**

A major point of contention is the absolute performance values reported.

Since PUM-Net explicitly utilizes an external memory corpus, it must be compared against standard RAG approaches to isolate the architectural benefit from the data access benefit.

Concerns regarding the evaluation protocol. The reliance on 130M parameter models for the majority of the ablation.

**Reviewer Scores:**

CVkt  and EZCr tend to remain rejection decisions.

---

### Decision · Program_Chairs · 2026-01-26

Reject